# OmniQuant: Omnidirectionally Calibrated Quantization for Large Language Models

**Wenqi Shao**[†1], **Mengzhao Chen**[†1], **Zhaoyang Zhang**[3], **Peng Xu**[1,2], **Lirui Zhao**[1],
**Zhiqian Li**[2], **Kaipeng Zhang**[1], **Peng Gao**[1], **Yu Qiao**[1], **Ping Luo**[*1,2]
[1]OpenGVLab, Shanghai AI Laboratory [2]The University of Hong Kong
[3]The Chinese University of Hong Kong

## Abstract

Large language models (LLMs) have revolutionized natural language processing tasks. However, their practical deployment is hindered by their immense memory and computation requirements. Although recent post-training quantization (PTQ) methods are effective in reducing memory footprint and improving the computational efficiency of LLM, they hand-craft quantization parameters, leading to low performance, especially in extremely low-bit quantization. To tackle this issue, we introduce an Omnidirectionally calibrated Quantization (**OmniQuant**) technique for LLMs, which achieves good performance in diverse quantization settings while maintaining the computational efficiency of PTQ by efficiently optimizing various quantization parameters. OmniQuant comprises two innovative components including Learnable Weight Clipping (LWC) and Learnable Equivalent Transformation (LET). LWC modulates the extreme values of weights by optimizing the clipping threshold. Meanwhile, LET tackles activation outliers by shifting the challenge of quantization from activations to weights. Operating within a differentiable framework using block-wise error minimization, OmniQuant can optimize the quantization process efficiently for both weight-only and weight-activation quantization. For instance, the LLaMA-2 model family size 7-70B can be processed with OmniQuant on a single A100-40G GPU within 1-16 hours using 128 samples. Extensive experiments validate OmniQuant's superior performance across diverse quantization configurations such as W4A4 (4-bit weight, 4-bit activation), W6A6, W4A16, W3A16, and W2A16. Additionally, OmniQuant demonstrates effectiveness in instruction-tuned models and delivers notable improvements in inference speed and memory reduction on real devices. Codes are available at `https://github.com/OpenGVLab/OmniQuant`.

## 1 Introduction

Large language models (LLMs) such as GPT-4 (Bubeck et al., 2023) and LLaMA (Touvron et al., 2023a), have demonstrated impressive performance across various natural language benchmarks (Hendrycks et al., 2020; Zellers et al., 2019). Furthermore, the language understanding capabilities inherent in LLMs can be successfully transferred into multimodal models (Mu et al., 2023; Xu et al., 2023; Zhang et al., 2023a; Huang et al., 2024; 2023). Thereby, LLMs can be regarded as precursors to artificial general intelligence (Bubeck et al., 2023). However, the considerable computational and memory requirements of LLMs pose substantial challenges (Zhang et al., 2023b; Hu et al., 2023). For instance, the GPT-3 model (Brown et al., 2020) requires 350G of memory to load its parameters in FP16 format, which corresponds to the requirement of at least five A100-80G GPUs for inference. This significant demand for computational resources and associated communication overheads impedes the practical deployment of LLMs in real-world applications.

Quantization has shown to be promising to mitigate both computational and memory overhead in LLMs. In general, it comes in two types including post-training quantization (PTQ) and quantization-aware training (QAT). Although QAT can lead to more competitive accuracy than PTQ,

---

[*]Corresponding author: Ping Luo, pluo@cs.hku.hk
[†] Equal Contribution

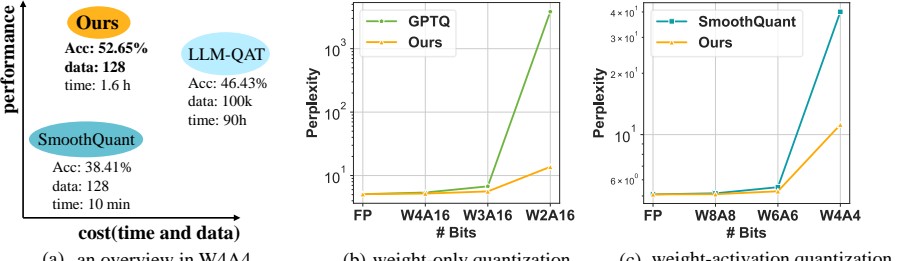

Figure 1: (a) provides an overview of LLaMA-7B with W4A4 quantization, highlighting Omni-Quant's ability to achieve quantization-aware training (QAT) performance with post-training quantization (PTQ) time and data efficiency. (b) and (c) showcase the perplexity (low is better) of quantized LLaMA-13B across different bit-widths on WikiText2.

it is not practical due to the high training cost because the whole model is trained with the awareness of the quantization process. As a result, PTQ is commonly utilized in existing quantization methods on LLMs. For example, lots of PTQ methods (Frantar et al., 2022; Lin et al., 2023; Dettmers et al., 2023b) reduce memory consumption by weight-only quantization which quantizes the weights while maintaining full-precision activation. To further reduce the computational overhead, another line of work (Xiao et al., 2023; Wei et al., 2022; Yuan et al., 2023; Wei et al., 2023; Liu et al., 2023a) employs weight-activation quantization which quantizes both weight and activation into low-bit values for the execution of low-bit matrix multiplication.

Existing quantization methods have demonstrated significant achievements in various scenarios, including W4A16 (*i.e.* 4-bit weight and 16-bit activation) weight-only quantization such as (Lin et al., 2023; Dettmers et al., 2023b; Lee et al., 2023), as well as W8A8 weight-activation quantization (Wei et al., 2023). However, they usually exhibit significant performance degradation when confronted with low-bit quantization, such as W2A16 and W4A4, as illustrated in Figure 1 (b & c). This performance shortfall in low-bit quantization can be attributed to the fact that these methods (Frantar et al., 2022; Lin et al., 2023; Wei et al., 2023) primarily rely on handcrafted quantization parameters such as migration strength (Xiao et al., 2023) and scaling parameters (Wei et al., 2023), which often leads to lower performance. Although Quantization-Aware Training (QAT) (Liu et al., 2023b) is effective in determining the optimal quantization configurations, it introduces substantial training overhead in both training and data efficiency. It is thus hard to quantize LLMs with QAT-based techniques efficiently such as LLMQAT (Liu et al., 2023b). For instance, GPTQ (Frantar et al., 2022), a PTQ approach, can complete the quantization of LLaMA-13B in an hour using 128 samples on a single A100 GPU, while LLM-QAT (Liu et al., 2023b) requires 100k samples and hundreds of GPU hours. This leads us to a central question: *can we attain the performance of QAT, while maintaining the time and data efficiency of PTQ?*

This paper introduces a novel quantization technique, OmniQuant, which effectively addresses the above question. OmniQuant achieves state-of-the-art performance across various quantization scenarios, particularly in low-bit settings, while preserving the time and data efficiency of PTQ, as illustrated in Figure 1. Unlike Quantization-Aware Training (QAT) (Liu et al., 2023b) which involves cumbersome weight optimization, OmniQuant freezes the original full-precision weight and only incorporates a few learnable quantization parameters. As shown in Figure 2, OmniQuant consists of two key components that incorporate different types of learnable quantization parameters, including Learnable Weight Clipping (LWC) and Learnable Equivalent Transformation (LET). Specifically, LWC modulates the extreme values of weights by optimizing the clipping threshold. In the meanwhile, LET tackles activation outliers by learning mathematically equivalent transformations in a transformer encoder.

Instead of jointly optimizing all parameters across the LLM, OmniQuant sequentially quantizes the parameters of one layer before moving on to the next under a block-wise quantization error minimization framework. In this way, OminiQuant can be optimized efficiently using a simple Stochastic Gradient Descent (SGD) algorithm. Thanks to the differentiable optimization, LWC and LET can be seamlessly integrated into the quantization. We find that LWC can mitigate the difficulty in quantizing weights and LET further shifts the challenge of quantization from activations to weights, facilitating OmniQuant a versatile quantization framework for both weight-only and weight-activation quantization. Notably, OmniQuant introduces no extra computation or parameters

for the quantized model because the clipping threshold in LWC and equivalent factors in LET can be fused into quantized weights.

As depicted in Figure 2, OmniQuant is easy to implement even with limited resources. Especially, taking the LLaMA-2 model family (7B-70B) as an example, all models can be quantized on a single A100-40G GPU utilizing only 128 training samples. The training time ranges from 1 to 16 hours, depending on the size of the quantized model, which ranges from 7B to 70B. Owing to the seamless integration of LWC and LET achieved by differentiable optimization, OmniQuant exhibits superior performance compared to prior PTQ-based methods in various quantization settings. For example, when

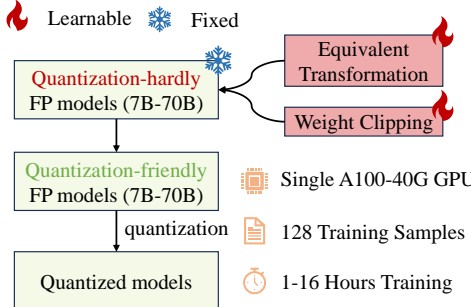

Figure 2: Characteristics of OmniQuant on LLaMA family.

LLaMA-13B is quantized into W2A16, OmniQuant achieves a perplexity of 13.21, while GPTQ incurs a significant increase in perplexity to 3832, as demonstrated in Figure 1. A similar performance advancement is also observed in the W4A4 quantization.

The contributions of **OmniQuant** are summarized as follows. 1) We formulate a novel quantization pipeline for LLM, OmniQuant, which freezes original full-precision weights while incorporating a restrained set of learnable parameters. OmniQuant imbues quantization with gradient updates while preserving the time and data efficiency of PTQ methods. 2) OmniQuant consists of Learnable Weight Clipping (LWC) and Learnable Equivalent Transformation (LET). These strategies make full-precision weights and activations more amenable to quantization. 3) Through extensive experiments, we demonstrate that OmniQuant outperforms previous methods across a spectrum of quantization settings (W416, W3A16, W2A16, W6A6, W4A4), various model families (OPT, LLaMA, LLaMA-2, LLaMA-2-chat, Falcon), and a range of model sizes (125M-180B). The computation speedup and memory reduction of OmniQuant are also demonstrated on real devices.

## 2 RELATED WORK

### 2.1 QUANTIZATION METHODS.

Quantization reduces neural network bit-precision, leading to smaller models and faster inference. Current methods are largely divided into Quantization Aware Training (QAT)(Liu et al., 2023b) and Post-training Quantization (PTQ)(Xiao et al., 2023; Frantar et al., 2022). While QAT maintains performance by simulating quantization during training, its training cost makes it unsuitable for LLM. PTQ techniques like AdaRound (Nagel et al., 2020) and BRECQ (Li et al., 2021) use gradient optimization to determine optimal rounding, but tuning all weights is time-intensive for larger models. Thus, most LLM quantization methods (Xiao et al., 2023; Frantar et al., 2022; Dettmers et al., 2023b; Lee et al., 2023; Wei et al., 2023) prioritize training-free PTQ, which limit performance in lower-bit situations. Our goal is to integrate gradient updates in LLM quantization, mirroring QAT's approach, while retaining PTQ's efficiency.

### 2.2 QUANTIZATION OF LLM.

Considering the quantized object, exiting LLM quantization can be classified into two fields: weight-only quantization and weight-activation quantization.

**Weight-only quantization.** Weight-only quantization focuses on converting weights to low-bit values. For instance, GPTQ (Frantar et al., 2022) uses block-wise reconstruction for 3/4-bit quantization. SpQR (Dettmers et al., 2023b), OWQ (Lee et al., 2023), and AWQ (Lin et al., 2023) emphasize the significance of weights tied to higher-magnitude activations. Therefore, SpQR and OWQ employ mixed-precision quantization to safeguard vital weights, while AWQ opts for channel-wise scaling to avoid mixed-precision's hardware inefficiency. Qlora (Dettmers et al., 2023a) and INT2.1 (Chee et al., 2023) restore the capabilities of the quantized model through parameter-efficient fine-tuning. Our method, in contrast, enhances the quantization process directly, making OmniQuant complementary to Qlora and INT2.1.

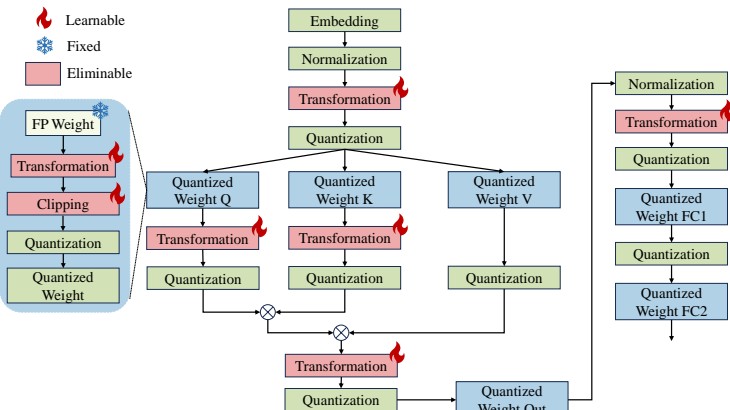

Figure 3: **Details of OmniQuant** in a transformer block. Note that all learnable parameters can be eliminated after quantization.

**Weight-activation quantization.** Weight-activation quantization compresses both weights and activations. SmoothQuant (Xiao et al., 2023), LLM.int8() (Dettmers et al., 2022), and Outlier Suppression (Wei et al., 2022) achieve W8A8 quantization by managing activation outliers. LLM.int8() uses mixed-precision decomposition, while the other two employ channel-wise scaling. Furthermore, Outlier Suppression+(Wei et al., 2023) adds channel-wise shifting to drive W6A6 quantization. Unlike previous heuristic designs, we use gradient optimization and expand equivalent transformations to attention mechanisms, further boosting the K/V cache quantization. Recently, RPTQ (Yuan et al., 2023) and LLM-QAT (Liu et al., 2023b) have achieved W4A4 quantization. However, RPTQ adopts deployment-unfriendly group-wise activation quantization, and LLM-QAT employs time-consuming QAT. In distinction from RPTQ and LLM-QAT, we achieve W4A4 quantization through deployment-friendly per-token quantization and maintain the PTQ efficiency.

## 3 OMNIQUANT

**Challenge of LLM quantization.** Two main difficulties lie in quantizing an LLM. First, the activation is hard to quantize due to the existence of outlier channels. Considering that weight distribution is flat and uniform, SmoothQuant (Xiao et al., 2023) and Outlier Suppression+ (Wei et al., 2023) tackle this issue by migrating the quantization difficulty from activations to weights with a pre-defined migration strength or grid-searching based optimization. Second, the quantization error of weights also plays a pivotal role in the final performance due to the importance of weights corresponding to activations. SqQR (Dettmers et al., 2023b) and OWQ (Lee et al., 2023) propose to retain crucial weights in full-precision, while AWQ (Lin et al., 2023) safeguards these weights using grid-searched channel-wise scaling. Although these methods have achieved certain success in compressing various LLMs, they often lead to suboptimal performance and fail to deal with extremely low-bit quantization due to the crude design of hand-crafted quantization parameters such as migration strength and scaling factors.

In this section, we introduce a differentiable quantization technique for LLM called **OmniQuant** where quantization parameters are learned with better flexibility. Towards this goal, OmniQuant is implemented with a block-wise quantization error minimization framework as presented in Sec.3.1. To tackle the aforementioned challenges of LLM quantization, we devise two novel strategies for additional learnable quantization parameters including a learnable weight clipping (LWC) to mitigate the difficulty in quantizing weights and a learnable equivalent transformation (LET) to further shift the challenge of quantization from activations to weights. We introduce LWC and LCT in Sec. 3.2 and Sec. 3.3, respectively.

### 3.1 BLOCK-WISE QUANTIZATION ERROR MINIMIZATION

Previous PTQ methods with gradient optimization, such as AdaRound (Nagel et al., 2020), BRECQ (Li et al., 2021) cannot be applied in models with billions of parameters because they are hard to optimize due to the huge solution space. Instead of turning the whole model, we propose a new optimization pipeline with block-wise quantization error minimization where the additional

quantization parameters can be optimized in a differentiable manner. We formulate the optimization goal as follows:

$$\arg\min_{\Theta_1,\Theta_2} ||\mathcal{F}(\mathbf{W}, \mathbf{X}) - \mathcal{F}(Q_w(\mathbf{W}; \Theta_1, \Theta_2), Q_a(\mathbf{X}, \Theta_2))||, \tag{1}$$

where $\mathcal{F}$ represents the mapping function for a transformer block in the LLM, $\mathbf{W}$ and $\mathbf{X}$ are full-precision weight and activation, $Q_w(\cdot)$ and $Q_a(\cdot)$ represent weight and activation quantizer, respectively, $\Theta_1$ and $\Theta_2$ are quantization parameters in learnable weight clipping (LWC) and learnable equivalent transformation (LET), respectively. The Block-wise quantization in Eqn.(1) sequentially quantizes the parameters of one transformer block before moving on to the next.

Block-wise minimization in Eqn.(1) has two advantages. First, equipped with block-wise minimization in Eqn.(1), OmniQuant can optimize quantization parameters in LWC and LET jointly, making it capable enough to encompass both weight-only and weight-activation quantization. Second, block-wise minimization is easy to optimize with minimal resource requirements. OmniQuant only determines a few quantization parameters with optimality, which is easier than optimizing the whole weights in previous PTQ-based methods (Nagel et al., 2020; Li et al., 2021). Empirically, we find that all models from the LLaMA-2 family (Touvron et al., 2023b) can be quantized on a single A100-40G GPU utilizing only 128 training samples.

## 3.2 LEARNABLE WEIGHT CLIPPING

OmniQuant employs a module of learnable weight clipping (LWC) to reduce the difficulty of quantizing the weights in an LLM. Similar to previous methods with learnable clipping threshold (Esser et al., 2019; Liu et al., 2022; Choi et al., 2018), LWC also determines the optimal dynamic range of the weights by optimizing a clipping threshold. However, we find that directly employing prior arts such as PACT (Choi et al., 2018) and LSQ (Esser et al., 2019) in quantization would produce unsatisfactory performance, as demonstrated in Table A14 in the Appendix.

Instead of directly learning a clipping threshold as in previous methods (Esser et al., 2019; Choi et al., 2018), LWC optimizes a clipping strength as formulated by

$$\mathbf{W_q} = \text{clamp}(\lfloor\frac{\mathbf{W}}{h}\rceil + z, 0, 2^N - 1), \text{where } h = \frac{\gamma\max(\mathbf{W}) - \beta\min(\mathbf{W})}{2^N - 1}, z = -\lfloor\frac{\beta\min(\mathbf{W})}{h}\rceil \tag{2}$$

where $\lfloor\cdot\rceil$ indicates round operation. $N$ is the target bit number. $\mathbf{W}_q$ and $\mathbf{W}$ denote the quantized and full-precision weights, respectively. $h$ is the normalization factor for weights and $z$ is the zero-point value. The clamp operation constrains the value within the range of $N$-bit integer, specifically $[0, 2^N - 1]$. In Eqn.(2), $\gamma \in [0, 1]$ and $\beta \in [0, 1]$ are learnable clipping strengths for the upper and the lower bound of weights, respectively. We instantiate $\gamma$ and $\beta$ by the sigmoid function[*]. Hence, $\Theta_1 = \{\gamma, \beta\}$ in Eqn.(1).

Note that LWC degrades into a vanilla MinMax quantization scheme used in existing works (Xiao et al., 2023),Frantar et al. (2022) when $\gamma = 1$ and $\beta = 1$. By inheriting the benefits of Min-Max quantization, LWC only needs to adjust the clipping strengths to determine an optimal clipping threshold, which would reduce the optimization difficulty. Clipped by an optimal threshold, the original weights would be easy to quantize. As indicated by the experiments in Table 1, our proposed learnable weight clipping method significantly outperforms previous weight-only quantization techniques (Frantar et al., 2022; Lin et al., 2023)).

## 3.3 LEARNABLE EQUIVALENT TRANSFORMATION

Other than LWC which enables quantization-friendly weights by optimizing the clipping threshold, we further reduce the difficulty of weight-activation quantization by a learnable equivalent transformation (LET). Considering that outliers in the activation map are systematic and unique to specific channels, previous methods such as SmoothQuant (Xiao et al., 2023) migrate the difficulty of quantization from activations to weights with a mathematically equivalent transformation. However, they hand-craft the equivalent parameters, leading to suboptimal results.

---

[*]Sigmoid$(t) = 1/(1 + \exp^{-t})$

Thanks to the inclusion of block-wise quantization error minimization, our LET can determine the optimal equivalent parameters in a differentiable way. Inspired by SmoothQuant (Xiao et al., 2023) and Outlier Suppression+ (Wei et al., 2023), we adopt channel-wise scaling and channel-wise shifting to manipulate the activation distribution, providing an effective solution for the outlier issue. Specifically, we investigate the equivalent transformation across both the linear layer and attention operation, as illustrated in Figure 3.

**Linear layer.** The linear layer takes an input token sequence $\mathbf{X} \in \mathbb{R}^{T \times C_{in}}$ where $T$ is the token length and is the multiplication of the weight matrix $\mathbf{W} \in \mathbb{R}^{C_{in} \times C_{out}}$ and bias vector $\mathbf{B} \in \mathbb{R}^{1 \times C_{out}}$. A mathematically equivalent linear layer is expressed as:

$$\mathbf{Y} = \mathbf{XW} + \mathbf{B} = [\underbrace{(\mathbf{X} - \delta) \oslash s}_{\tilde{\mathbf{X}}}] \cdot [\underbrace{s \odot \mathbf{W}}_{\tilde{\mathbf{W}}}] + [\underbrace{\mathbf{B} + \delta \mathbf{W}}_{\tilde{\mathbf{B}}}] \tag{3}$$

where $\mathbf{Y}$ represents the output, $\mathbf{s} \in \mathbb{R}^{1 \times C_{in}}$ and $\delta \in \mathbb{R}^{1 \times C_{in}}$ are channel-wise scaling and shifting parameters, respectively, $\tilde{\mathbf{X}}, \tilde{\mathbf{W}}$ and $\tilde{\mathbf{B}}$ are equivalent activation, weight and bias, respectively, '$\oslash$' and '$\odot$' are elementwise division and multiplication. By Eqn.(3), the activations are transformed to be quantization-friendly at a cost of increased quantization difficulty in weights. In this sense, LWC in Sec. 3.2 can improve the performance of weight-activation quantization achieved by LET because it renders weights quantization-friendly. Finally, we perform quantization on transformed activations and weights, as given by

$$\mathbf{Y} = Q_a(\tilde{\mathbf{X}})Q_w(\tilde{\mathbf{W}}) + \tilde{\mathbf{B}}, \tag{4}$$

where $Q_a$ is the vanilla MinMax quantizer and $Q_w$ is the MinMax quantizer with learnable weight clipping (i.e. our LWC).

Note that the scaling and shifting parameters in $\tilde{\mathbf{X}}$ can be absorbed into the previous normalization or linear layer and the the scaling factors in $\tilde{\mathbf{W}}$ can be fused into the original linear weight $\mathbf{W}$. Therefore, the equivalent transformation in Eqn.(3) can effectively reduce quantization errors without introducing additional parameters or costs. We employ this equivalent transformation in all linear layers of the LLM except for the second linear layer of FFN as shown in Figure 3. This may be because the high sparsity of features after the non-linear layer (Liu et al., 2023c) leads to unstable gradients when applying learnable equivalent transformations.

**Attention operation.** Beyond the linear layer, the attention operation also accounts for a significant proportion of the computation. Additionally, the auto-regressive pattern of LLM necessitates storing the key-value(KV) cache for each token, which results in substantial memory demands for long sequences. Therefore, we also quantize $\mathbf{Q}/\mathbf{K}/\mathbf{V}$ matrixes into low-bit in the weight-activation quantization setting. Specifically, the learnable equivalent transform of the self-attention affinity matrix can be written as:

$$\mathbf{P} = \text{Softmax}(\mathbf{Q}\mathbf{K}^T) = \text{Softmax}((\underbrace{\mathbf{Q} \oslash s_a}_{\tilde{\mathbf{Q}}})(\underbrace{s_a \odot \mathbf{K}^T}_{\tilde{\mathbf{K}}^T})). \tag{5}$$

where $s_a \in \mathbb{R}^{1 \times C_{out}}$ is the scaling factor in the affinity matrix. Similar to Eqn.(4), the quantized affinity matrix calculation is expressed as $\mathbf{P} = \text{Softmax}(Q_a(\tilde{\mathbf{Q}})Q_a(\tilde{\mathbf{K}}^T))$. Here we also use Min-Max quantization scheme as $Q_a$ to quantize $\tilde{\mathbf{Q}}/\tilde{\mathbf{K}}$ matrixes. From Eqn.(4) and Eqn.(5) we know that $\Theta_2 = \{\delta, s, s_a\}$ in Eqn.(1).

The channel-wise scaling factors in $\tilde{\mathbf{Q}}$ and $\tilde{\mathbf{K}}$, as seen in Eq.(5), can be absorbed into linear weights of the query and key projection, respectively. It is worth mentioning that the explicit transformation of $\mathbf{V}$ is omitted as its distribution has already been channel-wise altered by the inverse transformation associated with the output projection linear layer.

## 4 EXPERIMENTS

### 4.1 SETTINGS

**Quantization.** We experiment with both weight-only and weight-activation quantization. For the former, default settings are INT4/INT3/INT2 per-channel weight quantization. Group-wise weight

Table 1: **Weight-only quantization Results of LLaMA-1 and LLaMA-2 Models**. We report WikiText2 perplexity in this table, C4 perplexity can be found in Table A19 in Appendix.

| LLaMA1&2 / PPL↓ | | 1-7B | 1-13B | 1-30B | 1-65B | 2-7B | 2-13B | 2-70B |
|---|---|---|---|---|---|---|---|---|
| FP16 | - | 5.68 | 5.09 | 4.10 | 3.53 | 5.47 | 4.88 | 3.31 |
| W2A16 | RTN | 1.1e5 | 6.8e4 | 2.4e4 | 2.2e4 | 3.8e4 | 5.6e4 | 2.0e4 |
| | GPTQ | 2.1e3 | 5.5e3 | 499.75 | 55.91 | 7.7e3 | 2.1e3 | 77.95 |
| | **OmniQuant** | **15.47** | **13.21** | **8.71** | **7.58** | **37.37** | **17.21** | **7.81** |
| W2A16 g128 | RTN | 1.9e3 | 781.20 | 68.04 | 15.08 | 4.2e3 | 122.08 | 27.27 |
| | GPTQ | 44.01 | 15.60 | 10.92 | 9.51 | 36.77 | 28.14 | NAN |
| | AWQ | 2.6e5 | 2.8e5 | 2.4e5 | 7.4e4 | 2.2e5 | 1.2e5 | - |
| | **OmniQuant** | **9.72** | **7.93** | **7.12** | **5.95** | **11.06** | **8.26** | **6.55** |
| W2A16 g64 | RTN | 188.32 | 101.87 | 19.20 | 9.39 | 431.97 | 26.22 | 10.31 |
| | GPTQ | 22.10 | 10.06 | 8.54 | 8.31 | 20.85 | 22.44 | NAN |
| | AWQ | 2.5e5 | 2.7e5 | 2.3e5 | 7.4e4 | 2.1e5 | 1.2e5 | - |
| | **OmniQuant** | **8.90** | **7.34** | **6.59** | **5.65** | **9.62** | **7.56** | **6.11** |
| W3A16 | RTN | 25.73 | 11.39 | 14.95 | 10.68 | 539.48 | 10.68 | 7.52 |
| | GPTQ | 8.06 | 6.76 | 5.84 | 5.06 | 8.37 | 6.44 | 4.82 |
| | AWQ | 11.88 | 7.45 | 10.07 | 5.21 | 24.00 | 10.45 | - |
| | **OmniQuant** | **6.49** | **5.68** | **4.74** | **4.04** | **6.58** | **5.58** | **3.92** |
| W3A16 g128 | RTN | 7.01 | 5.88 | 4.87 | 4.24 | 6.66 | 5.51 | 3.97 |
| | GPTQ | 6.55 | 5.62 | 4.80 | 4.17 | 6.29 | 5.42 | 3.85 |
| | AWQ | 6.46 | 5.51 | 4.63 | 3.99 | 6.24 | 5.32 | - |
| | **OmniQuant** | **6.15** | **5.44** | **4.56** | **3.94** | **6.03** | **5.28** | **3.78** |
| W4A16 | RTN | 6.43 | 5.55 | 4.57 | 3.87 | 6.11 | 5.20 | 3.67 |
| | GPTQ | 6.13 | 5.40 | 4.48 | 3.83 | 5.83 | 5.13 | 3.58 |
| | AWQ | 6.08 | 5.34 | 4.39 | 3.76 | 6.15 | 5.12 | - |
| | **OmniQuant** | **5.86** | **5.21** | **4.25** | **3.71** | **5.74** | **5.02** | **3.47** |
| W4A16 g128 | RTN | 5.96 | 5.25 | 4.23 | 3.67 | 5.72 | 4.98 | 3.46 |
| | GPTQ | 5.85 | 5.20 | 4.23 | 3.65 | 5.61 | 4.98 | 3.42 |
| | AWQ | 5.81 | 5.20 | 4.21 | 3.62 | 5.62 | 4.97 | - |
| | **OmniQuant** | **5.77** | **5.17** | **4.19** | **3.62** | **5.58** | **4.95** | **3.40** |

quantization is represented by 'g', e.g., W3A16g128 means 3-bit weight-only quantization with a 128-group size. In weight-activation quantization, defaults are INT6/INT4 per-channel weight and per-token activation quantization (Dettmers et al., 2022). All intermediate activations are quantized into low-bit, excluding the SoftMax output, kept at full precision due to its long-tail distribution making it unsuitable for uniform quantization.

**Training** The channel-wise scaling factor is initialized with SmoothQuant (Xiao et al., 2023), and the channel-wise shifting factor is initialized using Outlier Suppression+ (Wei et al., 2023). To optimize the learnable parameters, we utilize the AdamW optimizer with zero weight decay. The learning rate for learnable weight clipping and equivalent transformation is set as $5e-3$ and $1e-2$, respectively. We employ a calibration dataset consisting of 128 randomly selected 2048-token segments from WikiText2 (Merity et al., 2016). The entire training process is facilitated on a single Nvidia A100 GPU, using a batch size of 1 over 20 epochs, except for W2A16 quantization that leverages 40 epochs. For weight-activation quantization, both learnable weight clipping and equivalent transformation are activated. For weight-only, both are used for OPT, but only the clipping is for LLaMA, as Table A3 shows negligible benefits from the equivalent transformation for LLaMA.

**Models.** We test on OPT(125M-66B)(Zhang et al., 2022)), LLaMA(7B-65B) (Touvron et al., 2023a), LLaMA-2(7B-70B) (Touvron et al., 2023b), Falcon-180B (Penedo et al., 2023), and instruction-tuned LLaMA-2-chat (Touvron et al., 2023b) for generalizability. While the main paper highlights the LLaMA results, comprehensive details for other models are available in Sec. A8 of the Appendix.

**Evaluation.** Following the previous work (Lin et al., 2023; Frantar et al., 2022), we evaluate quantized models by reporting the perplexity of language generation experiments, specifically on Wiki-Text2 (Merity et al., 2016), PTB (Marcus et al., 1994)), C4 (Raffel et al., 2020). Moreover, accuracy is evaluated in zero-shot tasks including PIQA (Bisk et al., 2020), ARC (Clark et al., 2018), BoolQ (Clark et al., 2019), and HellaSwag (Clark et al., 2018). We adhere to the GPTQ (Frantar et al., 2022) settings for language generation experiments, and implement the lm-eval-harness (Gao et al., 2021) for the execution of all zero-shot tasks.

**Baselines.** For weight-only quantization, we compare with vanilla round-to-nearest quantization (RTN), GPTQ (Frantar et al., 2022), and AWQ (Lin et al., 2023). For weight-activation quantization, we compare our method with SmoothQuant (Xiao et al., 2023), Outlier Supression + (Wei et al., 2023), RPTQ (Yuan et al., 2023), and the recent QAT method LLM-QAT (Liu et al., 2023b). Note

Table 2: **Weight-activation quantization results of LLaMA Models.** This table reports the accuracy of 6 zero-shot tasks. Perplexity results can be found in Table A23 & A24 at Appendix.

| LLaMA / Acc↑ | #Bits | Method | PIQA | ARC-e | Arc-c | BoolQ | HellaSwag | Winogrande | Avg. |
|---|---|---|---|---|---|---|---|---|---|
| LLaMA-1-7B | FP16 | - | 77.47 | 52.48 | 41.46 | 73.08 | 73.00 | 67.07 | 64.09 |
| | W6A6 | SmoothQuant | 76.75 | 51.64 | 39.88 | 71.75 | 71.67 | 65.03 | 62.81 |
| | W6A6 | OS+ | 76.82 | 51.35 | 41.13 | 72.08 | 71.42 | 65.98 | 61.13 |
| | W6A6 | **OmniQuant** | 77.09 | 51.89 | 40.87 | 72.53 | 71.61 | 65.03 | **63.17** |
| | W4A4 | SmoothQuant | 49.80 | 30.40 | 25.80 | 49.10 | 27.40 | 48.00 | 38.41 |
| | W4A4 | LLM-QAT | 51.50 | 27.90 | 23.90 | 61.30 | 31.10 | 51.90 | 41.27 |
| | W4A4 | LLM-QAT+SQ | 55.90 | 35.50 | 26.40 | 62.40 | 47.80 | 50.60 | 46.43 |
| | W4A4 | OS+ | 62.73 | 39.98 | 30.29 | 60.21 | 44.39 | 52.96 | 48.43 |
| | W4A4 | **OmniQuant** | 66.15 | 45.20 | 31.14 | 63.51 | 56.44 | 53.43 | **52.65** |
| LLaMA-1-13B | FP16 | - | 79.10 | 59.89 | 44.45 | 68.01 | 76.21 | 70.31 | 66.33 |
| | W6A6 | SmoothQuant | 77.91 | 56.60 | 42.40 | 64.95 | 75.36 | 69.36 | 64.43 |
| | W6A6 | OS+ | 78.29 | 56.90 | 43.09 | 66.98 | 75.09 | 69.22 | 64.92 |
| | W6A6 | **OmniQuant** | 78.40 | 57.28 | 42.91 | 67.00 | 75.82 | 68.27 | **64.95** |
| | W4A4 | SmoothQuant | 61.04 | 39.18 | 30.80 | 61.80 | 52.29 | 51.06 | 49.36 |
| | W4A4 | OS+ | 63.00 | 40.32 | 30.38 | 60.34 | 53.61 | 51.54 | 49.86 |
| | W4A4 | **OmniQuant** | 69.69 | 47.39 | 33.10 | 62.84 | 58.96 | 55.80 | **54.37** |
| LLaMA-1-30B | FP16 | - | 80.08 | 58.92 | 45.47 | 68.44 | 79.21 | 72.53 | 67.44 |
| | W6A6 | SmoothQuant | 77.14 | 57.61 | 42.91 | 65.56 | 78.07 | 69.92 | 65.20 |
| | W6A6 | OS+ | 80.14 | 58.92 | 45.05 | 68.02 | 77.96 | 71.98 | 67.01 |
| | W6A6 | **OmniQuant** | 79.81 | 58.79 | 45.22 | 68.38 | 78.95 | 72.21 | **67.23** |
| | W4A4 | SmoothQuant | 58.65 | 35.53 | 27.73 | 60.42 | 35.56 | 48.06 | 44.83 |
| | W4A4 | OS+ | 67.63 | 46.17 | 34.40 | 60.70 | 54.32 | 52.64 | 52.62 |
| | W4A4 | **OmniQuant** | 71.21 | 49.45 | 34.47 | 65.33 | 64.65 | 59.19 | **56.63** |
| LLaMA-1-65B | FP16 | - | 80.79 | 58.71 | 46.24 | 82.29 | 80.72 | 77.50 | 71.04 |
| | W6A6 | SmoothQuant | 80.25 | 57.92 | 45.50 | 80.22 | 80.18 | 74.76 | 69.80 |
| | W6A6 | OS+ | 79.67 | 55.68 | 45.22 | 80.02 | 78.03 | 73.95 | 68.76 |
| | W6A6 | **OmniQuant** | 81.01 | 58.12 | 46.33 | 80.64 | 79.91 | 75.69 | **70.28** |
| | W4A4 | SmoothQuant | 64.47 | 40.44 | 29.82 | 59.38 | 39.90 | 52.24 | 47.71 |
| | W4A4 | OS+ | 68.06 | 43.98 | 35.32 | 62.75 | 50.73 | 54.30 | 52.52 |
| | W4A4 | **OmniQuant** | 71.81 | 48.02 | 35.92 | 73.27 | 66.81 | 59.51 | **59.22** |

that we reproduce SmoothQuant and Outlier Suppression+ with per-channel weight quantization and per-token activation quantization for fair comparisons.

## 4.2 WEIGHT-ONLY QUANTIZATION RESULTS

The results of the LLaMA family can be found in Table 1, while the results for OPT are presented in the Sec. A8 of Appendix. As illustrated by the tables, OmniQuant consistently outperforms the prior LLM weight-only quantization method across various LLM families (OPT, LLaMA-1, LLaMA-2) and diverse quantization configurations, including W2A16, W2A16g128, W2A16g64, W3A16, W3A16g128, W4A16, and W4A16g128. These findings suggest OmniQuant's versatility, being adaptable to a multitude of quantization configurations. For instance, while AWQ (Lin et al., 2023) is particularly effective with group-wise quantization, OmniQuant demonstrates superior performance across both channel-wise and group-wise quantization. Furthermore, the performance benefits of OmniQuant become more pronounced as the quantization bit size decreases.

## 4.3 WEIGHT-ACTIVATION QUANTIZATION RESULTS

In weight-activation quantization, our main focus lies on W6A6 and W4A4 quantization. We exclude W8A8 quantization as SmoothQuant can nearly achieve lossless W8A8 quantized models when compared with full-precision counterparts. The results of the LLaMA family can be found in Table 2, while the results for OPT are presented in Table A25 of Appendix. Table 2 illustrates the zero-shot task accuracy of LLaMA weight-activation quantization. Notably, OmniQuant markedly enhances the average accuracy by +4.99% ∼ +11.80% across various models at W4A4 quantization. Remarkably, in the LLaMA-7B, OmniQuant even surpasses the recent QAT method, LLM-QAT (Liu et al., 2023b), by an impressive margin of +6.22%. This improvement demonstrates the efficacy of incorporating additional learnable parameters, which proves to be more beneficial than the global weight tuning utilized by QAT.

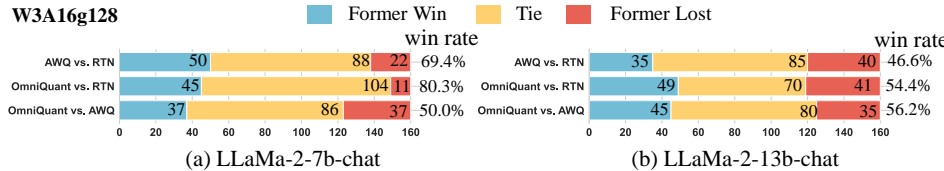

Figure 4: Comparing W3A16g128 quantization among RTN, AWQ (Lin et al., 2023), and Omni-Quant under Vicuna-Bench (Chiang et al., 2023). Win rates are calculated without considering tie samples. A higher win rate indicates the better performance of the former of *vs.* pairs.

Table 3: Deployment of weight-only quantization through MLC-LLM. We report the memory size of quantized weights (denoted as 'WM') and the running memory (denoted as 'RM') and speed in NVIDIA A100-80G.

| LLaMA | 7B | | | 13B | | | 30B | | | 65B | | |
|---|---|---|---|---|---|---|---|---|---|---|---|---|
| | WM | RM | token/s | WM | RM | token/s | WM | RM | token/s | WM | RM | token/s |
| FP | 12.6G | 14.4G | 69.2 | 24.3G | 27.1G | 52.5 | 60.6G | 66.1G | 23.9 | OOM | - | - |
| W4A16g128 | 3.8G | 5.7G | 134.2 | 7.0G | 10.0G | 91.3 | 16.7G | 21.7G | 43.6 | 33.0G | 41.0G | 24.3 |
| W3A16g128 | 3.2G | 5.1G | 83.4 | 5.8G | 8.7G | 57.6 | 13.7G | 18.7G | 29.0 | 27.0G | 35.1G | 15.2 |
| W2A16g128 | 2.2G | 4.1G | 83.9 | 4.0G | 7.5G | 92.6 | 9.2G | 14.1G | 36.7 | 18.0G | 25.6G | 24.8 |

## 4.4 QUANTIZATION OF INSTRUCTION-TUNED MODELS

To validate the generalization capability of our method, we test the quantization on LLaMA-2-chat (Touvron et al., 2023b), an instruction-tuned model for chatbots. Using the GPT-4 evaluation protocol (Chiang et al., 2023), performance is assessed on the Vicuna benchmark (Chiang et al., 2023) comprising 80 questions. To negate position bias (Zheng et al., 2023), each pair is compared in both sequences, totaling 160 trials per comparison. Figure 4 compares RTN, AWQ (Lin et al., 2023), and OmniQuant. In LLaMA-2-7b-chat, OmniQuant matches AWQ with a 50% win rate but surpasses RTN more (80.3% vs. 69.4%). In LLaMA-2-13b-chat, while AWQ lags behind RTN, OmniQuant consistently improves quantization model performance.

## 4.5 ACCELERATION ON REAL DEVICE

MLC-LLM[†] provides a versatile deployment solution for diverse language models across various hardwares. It particularly excels in deploying quantized models on CUDA. One of OmniQuant's strengths lies in its ability to avoid extra operations for quantized models, allowing MLC-LLM to seamlessly run models created with OmniQuant. Table,3 shows memory requirements and inference speeds of the LLaMA family on an NVIDIA A100-80G. 'Weights Memory (WM)' represents quantized weight storage, and 'Running Memory (RM)' indicates the memory for inference, with the latter being higher due to certain retained activations. Inference speed is gauged by generating 512 tokens. It is evident that quantized models significantly reduce memory usage compared to 16-bit full-precision models. For instance, models with W4A16g128 and W2A16g128 quantization almost double the inference speed. However, MLC-LLM's support for INT3/INT2 is currently suboptimal, particularly for INT3. Enhancements to INT3/INT2 quantization speed are in our future roadmap. Additionally, we only explore the deployment of weight-only quantization in this study due to that W4A4 and W6A6 quantization methods lack out-of-the-box hardware support.

## 5 CONCLUSION

We present OmniQuant, a method advancing weight-only and weight-activation quantization to low-bit formats. OmniQuant's core principle is to retain original full-precision weights while adding learnable parameters. It uses learnable weight clipping and learnable equivalent transformation to optimize weight and activation for quantization. While incorporating gradient updates, OmniQuant maintains training efficiency comparable to existing PTQ methods. It outperforms current methods in language generation and zero-shot tasks and is suited for instruction-tuned LLMs. Additionally, OmniQuant also ensures hardware compatibility as its added parameters can be absorbed.

---

[†]https://github.com/mlc-ai/mlc-llm

ACKNOWLEDGMENTS

This paper is partially supported by the National Key R&D Program of China No.2022ZD0161000 and the General Research Fund of Hong Kong No.17200622. We thank Wentao Liu from SenseTime for his valuable insights and discussions regarding LLM deployment. We also acknowledge Siyuan Feng from Apache TVM for assisting in the successful deployment of our OmniQuant in the MLC LLM project.

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

In this appendix, we provide further details as follows:

- Sec.A1: Presents the pseudo code for our OmniQuant algorithm.
- Sec.A2: Summarizes the distinctions with existing equivalent transformation methods.
- Sec.A3: Details ablation studies, encompassing the efficacy of each component, design choices for the learnable equivalent transformation, training time, and calibration data, etc.
- Sec.A4: Provides the detailed training time for the LLaMA family.
- Sec.A5: Explores the internal mechanisms of the proposed method.
- Sec.A6: Compares the proposed LWC with other clipping-based quantization approaches.
- Sec.A8: Showcases the complete results for OPT, LLaMA-1, LLaMA-2, and Falcon models.

## A1 OVERALL ALGORITHM

The comprehensive training algorithm of OmniQuant is illustrated in Algorithm 1. We employ a block-wise calibration strategy comprising three steps: initialization of learnable parameters (Lines 4-5), training these learnable parameters (Lines 6-15), transforming the model with learned parameters, and then quantization(Lines 16-18). The OmniQuant algorithm finds the optimal transformation to enhance the quantization compatibility of the LLM model. Additionally, due to the elegant design, OmniQuant can achieve rapid convergence using a small calibration dataset.

---

**Algorithm 1** Overall algorithm of OmniQuant.

**Input**: calibration dataset $\mathbf{X}$, pre-trained LLM model $\mathcal{M}$
**Output**: quantized model.

1: $\mathbf{X}_{fp} = \mathbf{X}_q = \mathbf{X}$     ▷ init inputs of full-precision and quantized models.
2: **for** $\mathcal{B}_i$ in $\mathcal{M}$ **do**:     ▷ block-wise calibration
3:     $\mathbf{X}_{fp} = \mathcal{B}_i(\mathbf{X}_{fp})$     ▷ update the input of full-precision model
4:     init learnable weight clipping parameters $\Theta_1$
5:     init learnable equivalent transformation $\Theta_2$
6:     **for** k in epochs **do**:
7:         **for** $(\mathbf{x}_q, \mathbf{x}_{fp})$ in $(\mathbf{X}_q, \mathbf{X}_{fp})$ **do**
8:             $\mathcal{B}_i' = \text{LET}(\mathcal{B}_i, \Theta_2)$     ▷ With Eq.(3),Eq.(5)
9:             $\mathcal{B}_i' = \text{Quantization\_with\_LWC}(\mathcal{B}_i', \Theta_1)$     ▷ With Eq.(2)
10:             $\mathbf{x}_q' = \mathcal{B}_i'(\mathbf{x}_q)$
11:             $\text{loss} = ||\mathbf{x}_{fp} - \mathbf{x}_q'||^2$     ▷ With Eq.(1)
12:             loss.backward()
13:             update $\Theta_1$ and $\Theta_2$ through gradient
14:         **end for**
15:     **end for**
16:     $\mathcal{B}_i = \text{LET}(\mathcal{B}_i, \Theta_2)$
17:     $\mathcal{B}_i = \text{Quantization\_with\_LWC}(\mathcal{B}_i, \Theta_1)$     ▷ obtain the quantized block
18:     $\mathbf{X}_q = \mathcal{B}_i(\mathbf{X}_q)$     ▷ update the input of quantized model
19: **end for**
20: **return** quantized model $\mathcal{M}$

---

Table A1: **Distinction of existing equivalent transformation methods.**

| Method | ET operation | ET position | ET parameters | application |
|---|---|---|---|---|
| SmoothQuant | scaling | linear layer | pre-defining | weight-activation quantization |
| AWQ | scaling | linear layer | grid searching | weight-only quantization |
| OP+ | scaling & shifting | linear layer | grid searching for scaling and pre-defining for shifting | weight-activation quantization |
| OmniQuant | scaling & shifting | linear layer **& attention** | **gradient-based optimization** | **weight-only quantization** & **weight-activation quantization** |

## A2  DISTINCTION OF EXISTING EQUIVALENT TRANSFORMATION METHODS

Equivalent transformation is popular in the quantization of large language models. In this section, we summarize the distinction of proposed OmniQuant with existing equivalent transformation works, including SmoothQuant (Xiao et al., 2023), AWQ Lin et al. (2023), Outlier Supression (OP+)+ Wei et al. (2023). As shown in Table A1:

- For the equivalent transformation operation, both SmoothQuant and AWQ only consider channel-wise scaling operation, while OP+ and OmniQuant consider both channel-wise scaling and shifting operation.

- For the execution position, previous methods only carry equivalent transformation on linear layers (Eq.(4)), while OmniQuant also considers the matrix multiplication within attention (Eq.(5)). This point enlarges the solution space of equivalent transformation and facilitates the quantization of $\mathbf{Q}$ and $\mathbf{K}$.

- For the manners to obtain parameters of equivalent transformation, SmoothQuant leverage pre-defined migration strength. Then, AWQ and OP+ introduce grid searching based on some heuristic proxy. However, OmniQuant optimized all equivalent transformation parameters through end-to-end gradient descent, which significantly improve the performance.

- For the application scenario, previous methods are designed for weight-only quantization or weight-activation quantization. However, because of the elegant design and cooperation of the proposed LWC and LET, OmniQuant can achieve excel in both weight-only quantization and weight-activation quantization.

## A3  ABLATION STUDIES

Table A2: **Effect of combination of equivalent transformation and weight clipping**. We report the average perplexity of WikiText2 and C4, and the average accuracy on 6 zero-shot tasks like Table 2.

| LLaMa-7B W4A4 | Average PPL ↓ | Average Acc. ↑ |
|---|---|---|
| SmoothQuant | 28.78 | 38.41 |
| LET | 16.97 | 48.83 |
| LET + grid-searched WC | 15.82 | 49.59 |
| SmoothQuant + LWC | 15.80 | 50.15 |
| **LET + LWC** | **12.87** | **52.65** |

**Combination of equivalent transformation and weight clipping.** The synergy between LET and LWC is achieved through a sophisticated differentiable framework as demonstrated in Algorithm 1, not a simple additive combination. LET performs activation-to-weight migration, and LWC further facilitates the quantization of weights, resulting in a seamless integration of the two techniques. In Table A2, we also test other combination variants, including replacing LET with SmoothQuant or replacing LWC with grid-searched weight clipping. The results show that training LET and LWC simultaneously achieves the best performance.

Table A3: **Efficacy of each component.** WikiText2 perplexity1 is reported in this table. '-' indicats remove the corresponding module from the overall proposed methods.

| PPL↓ | | LLaMA-13B | | OPT-13B | |
|---|---|---|---|---|---|
| **Method** | | W4A4 | W3A16 | W4A4 | W3A16 |
| | **LWC+LET** | **10.87** | **5.65** | **11.65** | **10.87** |
| | -LWC | 20.75 | 7.65 | 15.23 | 12.98 |
| components | -LET | 5.4e3 | 5.68 | 7.8e3 | 11.29 |
| | -LWC-LET | 1.8e3 | 10.68 | 7.8e5 | 4.6e3 |

**Efficacy of each component.** Table A3 reveals that the baseline model incorporates both LWC and LET, labeled as 'LWC+LET'. We further investigate their contributions by removing each component. Both components positively influence performance, but LET proves essential for weight-activation quantization. Disabling it for W4A4 results in a marked increase in perplexity to $e3$, mainly due to challenges with activation quantization outliers. For weight-only quantization, LET significantly boosts OPT's performance but offers a slight enhancement for LLaMA, explained by LLaMA's few weight outliers. For example, in naive W3A16 quantization (-LWC-LET), LLaMA reaches a perplexity of 10.68, while OPT's spikes to $4.6e3$. Consequently, LET is turned off for LLaMA in weight-only quantization given its limited advantage for faster training.

Table A4: **Design choices of learnable equivalent transformation.** WikiText2 perplexity1 is reported in this table.

| | PPL↓ | LLaMA-13B | | OPT-13B | |
|---|---|---|---|---|---|
| | **Method** | W4A4 | W3A16 | W4A4 | W3A16 |
| | **LWC+LET** | **10.87** | **5.65** | **11.65** | **10.87** |
| LET | -shifting | 11.47 | 5.65 | 13.64 | 10.87 |
| | -attention | 11.34 | 5.65 | 11.79 | 10.87 |

**Design choices of learnable equivalent transformation.** In comparison to the equivalent transformation incorporated in SmoothQuant (Xiao et al. (2023)), our approach additionally implements channel-wise shifting and attention transformation. The effects of these innovations are evaluated in Table A4. We can observe that both modifications enhance the performance of weight-activation quantization. However, the incremental benefit of the equivalent transformation in the attention operation is comparatively minor. This discrepancy is primarily due to the majority of outliers existing in the output of the normalization layer while being less prevalent in the $Q/K/V$ matrix.

Table A5: **Impact of LET on each position.** '-' indicates removing corresponding LET. We respectively remove the LET from each layer, and reporting the average perplexity of WikiText2 and C4, and the average accuracy on 6 zero-shot tasks like Table 2.

| LLaMa-7B | Average PPL ↓ | Average Acc. ↑ |
|---|---|---|
| **W4A4** | **12.87** | **52.65** |
| -[ln1, (q_proj, k_proj, v_proj)] | 19.87 | 46.79 |
| -[v_proj, out_proj] | 13.03 | 51.68 |
| -[Q,K] | 13.34 | 51.47 |
| -[ln2, fc1] | 14.47 | 51.04 |

**Impact of LET on each position.** We exclude the LET of the second linear layer due to the high sparsity of features after the non-linear layer leads to unstable gradients. Therefore, we have four LET pairs, represented as [ln1, (q_proj, k_proj, v_proj)], [v_proj, out_proj], [Q, K], and [ln2, fc1]. As shown in Table A5, we can find that all four LETs can improve the performance, specially for the [ln1, (q_proj, k_proj, v_proj)] pair. Such results also demonstrate that the activation outliers are more serious after layer normalization layers.

Table A6: **Impact of initialization of LET.** We report the average perplexity of WikiText2 and C4, and the average accuracy on 6 zero-shot tasks like Table 2.

| LLaMa-7B | Average PPL ↓ | Average Acc. ↑ |
|---|---|---|
| **W4A4** | **12.87** | **52.65** |
| initialize scaling as 1 | 13.64 | 51.37 |
| initialize shifting as 0 | 12.95 | 52.22 |

**Impact of initialization of LET.** We initialize the channel-wise scaling factor with SmoothQuant Xiao et al. (2023), and initialize the channel-wise shifting with Outlier Suppression+ Wei et al. (2023). To validate the impact of careful initialization, we try to initial scaling as 1 and initial shifting as 0. As shown in Table A6, we can find that careful initialization of scaling and shifting can improve the final performance. Specifically, scaling initialization is more important than shifting, since scaling plays the main role in alleviating outliers.

Table A7: **Impact of Softmax quantization.** We report the average perplexity of WikiText2 and C4, and the average accuracy on 6 zero-shot tasks like Table 2.

| LLaMa-7B | Average PPL ↓ | Average Acc. ↑ |
|---|---|---|
| **W4A4 + Softmax 16bit** | **12.87** | **52.65** |
| W4A4 + Softmax 8bit | 12.91 | 51.93 |
| W4A4 + Softmax 6bit | 13.20 | 51.70 |
| W4A4 + Softmax 4bit | 18.80 | 48.52 |

**Impact of Softmax quantization.** The output of SoftMax has a long-tailed distribution, making it unsuitable for uniform quantization. We carry out experiments to quantize the Softmax output into different bit numbers. As shown in the following table, we can find that quantizing the output of softmax into 8-bit and 6-bit bring acceptable performance degeneration, which demonstrates that block-wise calibration can compensate for the loss of 8-bit and 6-bit Softmax quantization. However, 4-bit Softmax quantization brings significantly performance loss, which requires further exploration and additional trick such as log2 quantization in RepQViT (Li et al., 2023). Note that we keep the output of SoftMax as 16-bit if no special instruction.

Table A8: **Impact of iterative training of LWC and LET.** We report the average perplexity of WikiText2 and C4, and the average accuracy on 6 zero-shot tasks like Table 2.

| LLaMa-7B W4A4 | Average PPL ↓ | Average Acc. ↑ |
|---|---|---|
| **simultaneously** | **12.87** | **52.65** |
| each iteration | 13.56 | 50.91 |
| each epoch | 13.51 | 52.06 |
| each epoch + double training epochs 4bit | 12.80 | 52.50 |

**Impact of iterative training.** In our approach, LWC and LET are trained simultaneously, and we have also explored an iterative training approach by iterations or epochs. The results, as presented in Table A8, clearly indicate that training LWC and LET simultaneously yields the best performance. This experiment demonstrates that the synergy between LET and LWC creates a progressive process, where both techniques reinforce each other rather than interfere. To further support this statement, we conducted an additional experiment (last row in Table A8), training LWC and LET iteratively with double training epochs. The results show that simultaneous training with 20 epochs achieves comparable performance to iterative training with 40 epochs. This demonstrates the effectiveness and efficiency of training LWC and LET simultaneously.

Table A9: **Ablation of training time.** We train LLaMA-7B with different quantization configuration on 128 2048-tokens segments from WikiText2 over various epochs. '0' indicates only initialization without fine-tuning. Wikitext perplexity is reported in this table.

| Epochs | W4A16 | W3A16 | W2A16 | W6A6 | W4A4 |
|---|---|---|---|---|---|
| 0 | 6.29 | 24.04 | 1.1e5 | 6.16 | 33.93 |
| 10 | 5.87 | 6.51 | 27.49 | 5.96 | 12.04 |
| 20 | 5.85 | 6.49 | 17.46 | 5.95 | 11.26 |
| 40 | 5.86 | 6.47 | 15.47 | 5.95 | 11.23 |
| 80 | - | - | 14.77 | - | - |

**Training Time** As illustrated in Table A9, LLaMA-7B was trained across various epochs to determine the optimal convergence time. Most quantization configurations converge within 20 epochs, with the exception of W2A16, which necessitates 80 epochs. Consequently, we establish a training epoch of 20 for all configurations, except for W2A16, for which we set it to 40 in consideration of the training time.

**Calibration Data** OmniQuant utilizes gradient optimization on constrained calibration datasets, sourced from WikiText2 and comprising 128 segments with 2048 tokens each. This prompts concerns about potential overfitting to the calibration dataset. To explore this, we evaluated the calibration dataset's influence using two other datasets: Pile (Gao et al. (2020)) and c4 (Raffel et al. (2020)). As depicted in Table A10, the variance in perplexity across diverse calibration datasets

Table A10: **Ablation of calibration dataset.**

| LLaMA-7B/**PPL**↓ | **W3A16** | | **W4A4** | |
|---|---|---|---|---|
| Calibration Dataset | WikiText2 | C4 | WikiText2 | C4 |
| WikiText2 | 6.47 | 8.19 | 11.23 | 14.61 |
| C4 | 6.67 | 8.13 | 12.17 | 14.24 |
| Pile | 6.69 | 8.17 | 12.04 | 14.22 |
| Varience | 0.009 | 0.0006 | 0.17 | 0.03 |

Table A11: **Ablation of sample number of calibration dataset.**

| LLaMA-7B/**PPL**↓ | **W3A16** | | **W4A4** | |
|---|---|---|---|---|
| Sample Number | WikiText2 | C4 | WikiText2 | C4 |
| 16 | 6.47 | **8.18** | 11.56 | 14.84 |
| 32 | 6.47 | 8.18 | 11.48 | 14.80 |
| 64 | 6.48 | 8.19 | 11.40 | **14.57** |
| 128 | 6.47 | 8.19 | **11.23** | 14.61 |
| 256 | **6.46** | 8.19 | 11.41 | 14.90 |

is marginal, fluctuating between 0.0006 and 0.17. This underscores OmniQuant's robustness concerning calibration set distribution. Furthermore, the data efficiency of OmniQuant was gauged by modulating the number of training samples, as presented in Table A11. Remarkably, OmniQuant converges with as few as 16 samples. Our selection of 128 samples aligns with established practices in prior works (Frantar et al. (2022); Lin et al. (2023)).

Table A12: Omniquant runtime on LLaMA family. The time correspond to training 128 2048-tokes segment over 20 epochs and a batch size of 1 on a single NVIDIA A100-80G.

| LLaMA | 7B | 13B | 30B | 65B |
|---|---|---|---|---|
| weight-only | 1.1h | 2.2h | 4.5h | 8.9h |
| weight-activation | 1.6h | 3.3h | 7.3h | 14.4h |

## A4 TRAINING TIME

As shown in Table A12, we report the training time of the proposed OmniQuant within the LLaMA family. Note that for LLaMA, we only activate learnable weight clipping for weight-only quantization. Therefore, the training time for weight-only quantization is shorter relative to weight-activation quantization, given the fewer learnable parameters involved. While our proposed method necessitates a training time that is approximately 5× greater than GPTQ, it remains markedly faster than QAT methods, which demand hundreds of GPU hours.

Table A13: $l_1$ **distance between quantized model and full-precision model.** $||\mathbf{W} - \mathbf{W_q}||$ indicates the average $l_1$ distance between quantized weight and full-precision weight. $||\mathbf{X} - \mathbf{X}_q||$ denotes the $l_1$ distance between the output of last transformer block.

| **LLaMA-7B / $l_1$ ↓** | $||\mathbf{W} - \mathbf{W_q}||$ | | $||\mathbf{X} - \mathbf{X}_q||$ | |
|---|---|---|---|---|
| quantization | w/o LWC | w/ LWC | w/o LWC | w/ LWC |
| W2A16g128 | 0.0089 | **0.0082** | 3.24 | **1.36** |
| W2A16g64 | 0.0098 | **0.0086** | 3.51 | **1.44** |
| W3A16 | 0.0062 | **0.0044** | 2.80 | **1.05** |
| W3A16g128 | 0.0042 | **0.0040** | 1.37 | **0.79** |
| W4A16 | 0.0028 | **0.0024** | 0.98 | **0.61** |
| W4A16g128 | 0.0020 | **0.0019** | 0.68 | **0.47** |

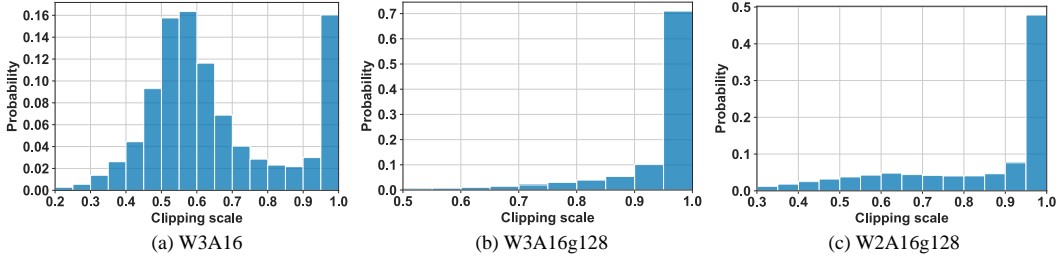

Figure A1: Visualization of learned clipping scale in different quantization settings in LLaMA-7B.

## A5 PERFORMANCE ANALYSIS

In this section, we investigate the internal mechanism of learnable weight clipping and learnable equivalent transformation respectively. Further, we show that with OmniQuant, 3-bit and 4-bit achieve similar trade-off between model bits and perplexity.

**Learnable weight clipping.** In addition to perplexity and accuracy, the quality of a quantization method can intuitively be evaluated by calculating the distance between quantized models and their full-precision counterparts. This is demonstrated in Table A13, where we detail the $l_1$ distance of weights and activations for LLaMA-7B's weight-only quantization. We can observe that the proposed Learned Weight Clipping (LWC) substantially decreases the $l_1$ distance for both weights and activations. It's noteworthy that, in certain instances, the $l_1$ distance for quantized models without LWC is similar to that of those utilizing LWC. However, models incorporating LWC exhibit markedly lower activation $l_1$ distances. This observation underpins the argument that LWC can effectively balance quantization precision between outlier and regular values.

Additionally, we illustrate the distribution of the learned clipping scale ($\gamma$ and $\beta$) as delineated in Eq. (2) in Figure A1. It is apparent that LWC can learn different clippings for diverse quantization configurations. For instance, with per-channel weight quantization W3A16 as depicted in Figure A1(a), the learned clipping scale showcases a normal distribution. This suggests that approximately half of the outliers are being clipped. In the case of group-wise quantization, the learned clipping scale exhibits a long-tailed distribution, implying that most quantized groups are associated with minimal clipping. Note that lower bits exhibit more pronounced clipping. For example, W2A16g128 possesses a 50% clipping scale larger than 0.95, whereas, in W3A16g128, this percentage rises to 70%.

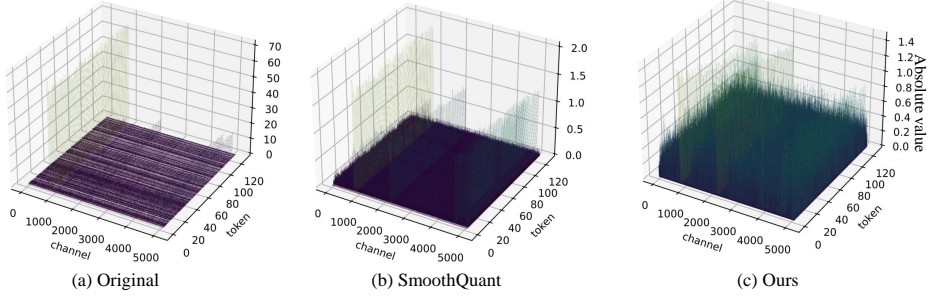

Figure A2: Visualization of activation of a linear layer in OPT-13B. (a) Original activation. (b) Activation after SmoothQuant. (c) Activation after proposed learnable equivalent transformation. Similar phenomena can be observed in different layer and different models.

**Learnable equivalent transformation.** Figure A2 provides visualizations of the intermediate activation in the linear layer. It is apparent that several outlier channels in the original activation (Figure A2(a)) possess significantly larger magnitudes compared to the regular channels, thereby creating an incompatibility with activation quantization. Although SmoothQuant mitigates this is-

sue to some degree, such as reducing the outlier magnitude from 70 to 2, Figure A2(b) reveals that the magnitude of outlier channels still remains notably larger than that of other regular channels after SmoothQuant. This phenomenon can be attributed to SmoothQuant's heuristic approach in deriving channel-wise scaling, which inevitably makes it challenging to discover an optimal solution. The impact of the proposed LET is depicted in Figure A2(c). It is noteworthy that the magnitude disparity between the outlier and regular channels is markedly diminished. This homogenization of the activation distribution, facilitated by the LET, empowers OmniQuant to efficiently steer the weight-activation quantization towards a low-bit scheme.

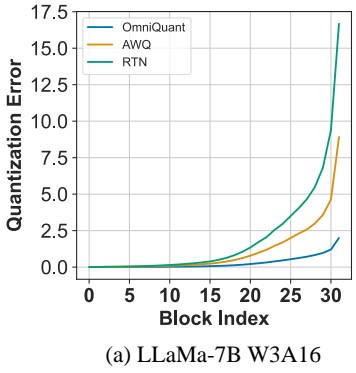 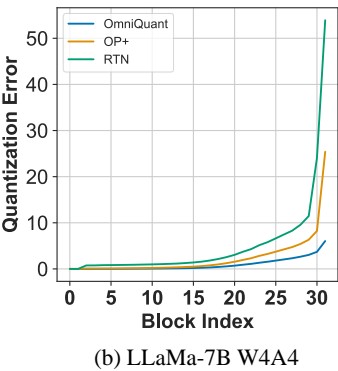

(a) LLaMa-7B W3A16            (b) LLaMa-7B W4A4

Figure A3: **Block-wise quantization error.** Grid-searched methods such as AWQ (Lin et al., 2023) and Outlier Suppression + (Wei et al., 2023) produce a more significant error than our gradient-based optimization method.

**Quantization error.** OmniQuant is the first differentiable post-training quantization algorithm for large language models. To demonstrate the advantage of gradient-based optimization, we also compare the quantization error of each block in Figure A3. We can find that OmniQuant significantly reduces the quantization loss compared with the grid-searching based method such as AWQ Lin et al. (2023) and Outlier Suppression + (Wei et al., 2023).

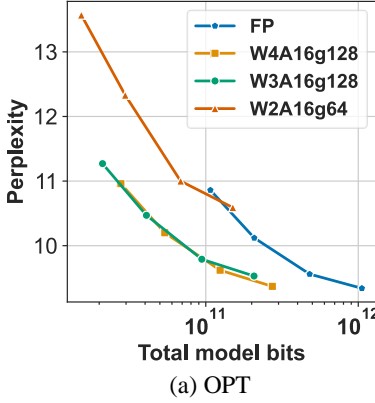 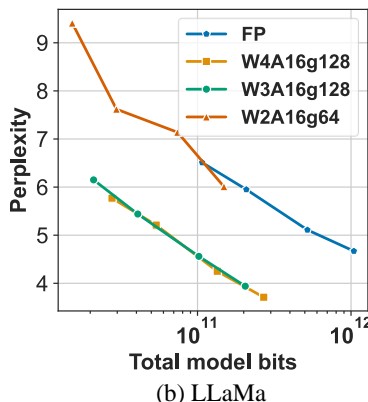

(a) OPT                 (b) LLaMa

Figure A4: Bit-level scaling laws for perplexity.

**Scaling laws.** Quantization serves as a potent strategy to curtail the total model bits, thereby facilitating the deployment of LLMs on edge or consumer devices with restricted memory. However, the total model bits are contingent on both the number of parameters within the original model and the quantization bits. Therefore, given a model bits constraint, the challenge arises: how does one optimally determine the number of parameters for the full-precision model and the quantization bits? Tim Dettmers (Dettmers & Zettlemoyer (2023)) demonstrated that 4-bit quantization establishes a universally optimal balance between the total model bits and zero-shot accuracy. Nonetheless, in this study, as shown in Figure A4, we would like to claim that OmniQuant can make 3-bit quantiza-

tion achieve comparable performance like 4-bit quantization in the trade off between model bits and perplexity.

Table A14: WikiText2 perplexity of clipping-based quantization methods. For fair comparison, we reproduce LSQ and PACT by replace LWC in our pipeline with them.

| LLaMA-7B/**PPL**↓ | Perplexity | |
|---|---|---|
| Method | W3A16 | W4A4 |
| FP | 5.68 | |
| MinMax | 25.73 | 14.49 |
| PACT (Choi et al. (2018)) | 6.95 | 18.25 |
| LSQ (Esser et al. (2019)) | 6.63 | 15.03 |
| LWC (Ours) | **6.47** | **11.26** |

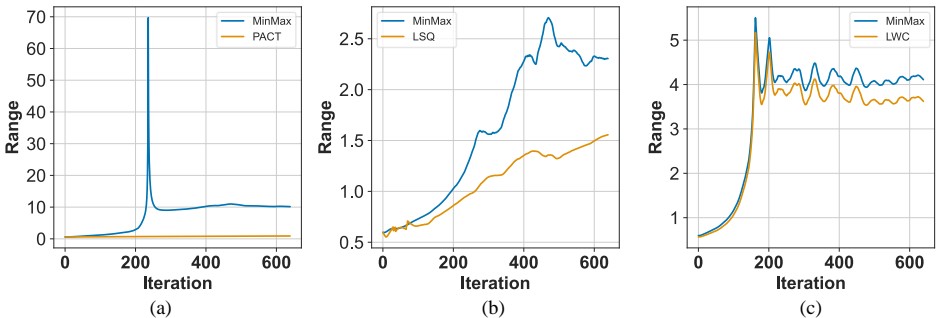

(a)          (b)          (c)

Figure A5: **Weights range changing of different clipping-based methods during training.** We plot the changing of weights range (maximum minus minimum) of the 3049-th output channel of the q-proj linear layer in the first LLaMa-1-7B block with W4A4 quantization. MinMax is the baseline which indicate withoud clipping. Similar phenomena can also be observed in other channels and other layers.

## A6   COMPARISONS WITH CLIPPING-BASED METHODS

In this paper, we proposed a novel method, learnable weight clipping (LWC), designed to adaptively determine the weight clipping threshold. LWC sets the threshold by scaling the original minimum and maximum values to delineate the solution space. We compare LWC against existing clipping-based methods: PACT and LSQ. While PACT directly determines the clipping threshold, LSQ focuses on the direct derivation of the scaling factor and zero-point. Both PACT and LSQ were initially formulated as QAT methods, accounting for both weight and activation clipping. For an equitable comparison, our examination is restricted to weight clipping. We integrated PACT and LSQ into our optimization pipeline in lieu of LWC. Table A14 illustrates that while PACT and LSQ enhance the performance of weight-only quantization compared to MinMax quantization, their efficacy diminishes in the weight-activation quantization setting. This decline can be attributed to the proposed LET during activation quantization, which alters the weight distribution in each training iteration, undermining the convergence of both LSQ and PACT. In contrast, LWC defines relative scaling values instead of absolute metrics, making it proficient in handling changes in weight distribution. For example, Figure A5 shows that LWC can catch the dramatically changing of weights while PACT and LSQ failed.

## A7   COMPARISONS WITH OTHER WEIGHT-ONLY QUANTIZATION METHODS

OmniQuant is an asymmetrically uniform quantization method. In the main paper, we compare with the same type of quantization methods, such as AWQ and GPTQ. Recently, there are also some other methods exploring for other quantization format. For example, SpQR (Dettmers et al., 2023b) and SqueezeLLM (Kim et al., 2023) employ mixed-precision quantization to safeguard vital weights.

Table A15: **Comparisons with SpQR and SqueezeLLM.**

| Size | Method | Avg bits | Wiki2 | C4 |
|---|---|---|---|---|
| LLaMa-1-7B | – | 16.00 | 5.68 | 7.08 |
| | SpQR | 3.94 | 5.87 | 7.28 |
| | SqueezeLLM | 4.07 | 5.79 | 7.20 |
| | SqueezeLLM | 4.27 | **5.77** | **7.18** |
| | OmniQuant | 4.16 | **5.77** | 7.21 |
| | SqueezeLLM | 3.05 | 6.20 | 7.67 |
| | SqueezeLLM | 3.24 | **6.13** | **7.56** |
| | OmniQuant | 3.15 | 6.15 | 7.75 |
| LLaMa-1-13B | – | 16.00 | 5.09 | 6.61 |
| | SpQR | 3.96 | 5.22 | 6.72 |
| | SqueezeLLM | 4.07 | **5.17** | 6.69 |
| | SqueezeLLM | 4.26 | **5.17** | **6.68** |
| | OmniQuant | 4.16 | **5.17** | 6.69 |
| | SqueezeLLM | 3.04 | 5.51 | 7.01 |
| | SqueezeLLM | 3.24 | 5.45 | **6.92** |
| | OmniQuant | 3.15 | **5.44** | 7.05 |
| LLaMa-1-30B | – | 16.00 | 4.10 | 5.98 |
| | SpQR | 3.89 | 4.25 | 6.08 |
| | SqueezeLLM | 4.06 | 4.20 | 6.05 |
| | SqueezeLLM | 4.25 | **4.18** | **6.04** |
| | OmniQuant | 4.16 | 4.19 | 6.06 |
| | SqueezeLLM | 3.04 | 4.56 | 6.31 |
| | SqueezeLLM | 3.24 | **4.44** | **6.23** |
| | OmniQuant | 3.15 | 4.56 | 6.37 |

Furthermore, SqueezeLLM also introduces non-uniform quantization to allocate more bits to sensitive weights. As shown in Table A15, we can find that OmniQuant can achieve comparable performance to SpQR and SqueezeLLM. While OmniQuant performs slightly worse than SqueezeLLM, our focus on uniform (INT) quantization provides simplicity and flexibility, supporting both weight-only quantization and weight-activation quantization. In contrast, SpQR and SqueezeLLM only support weight-only quantization. We believe this distinction adds valuable context to the comparison.

## A8 FULL RESULTS

In this section, we provide a comprehensive presentation of our results across various datasets to complement the main paper. Specifically, the results include:

- The perform overview (Figure A6).
- Experiments results on extreme large model Falcon-180B (Table A18).
- MMLU results on LLaMa-1-7B (Table A16).
- Asymmetric bits quantization, including W4A8 on LLaMa-1-7B, W4A6, and W8A4. (Table A17).
- C4 perplexity with weight-only quantization in the LLaMA families (Table A19).
- PTB perplexity with weight-only quantization in OPT families (Table A21).
- C4 perplexity with weight-only quantization in OPT families (Table A22).
- WikiText2 perplexity for weight-activation quantization in the LLaMA families (Table A23).
- C4 perplexity for weight-activation quantization in the LLaMA families (Table A24).

- WikiText2/PTB/C4 perplexity for weight-activation quantization in the LLaMA families (Table A25).

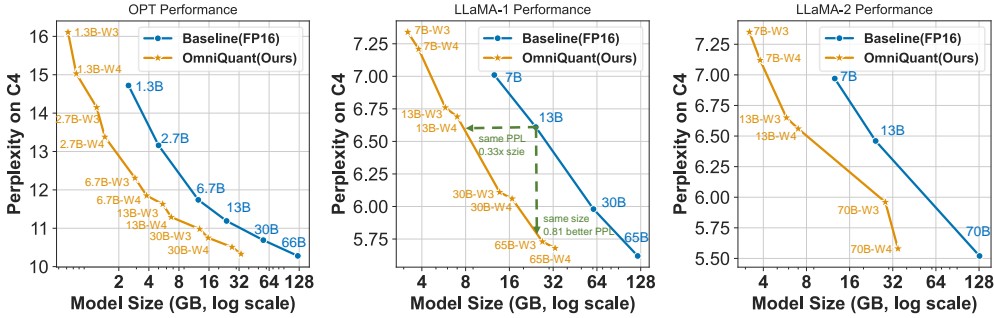

Figure A6: **Performance overview.** We display the trade-off curves for three model families. Each model showcases two quantization variants: W4A16g128 and W3A16g128. It is evident that Omni-Quant markedly enhances the trade-off between perplexity and model size. Specifically, OmniQuant delivers a reduction of 0.81 in perplexity for an equivalent model size and achieves the same perplexity with only 0.33x of the model size.

Table A16: **Average MMLU accuracy of LLaMa-7B.**

| LLaMa-1-7B (FP: 38.41%) | W4A16g128 | W3A16g128 | W2A16g128 | W4A4 |
|---|---|---|---|---|
| RTN | 37.37% | 33.43% | 22.55% | 23.31 |
| GPTQ | 35.39% | 30.53% | 23.83% | - |
| AWQ | **37.71%** | 35.43% | 22.58% | - |
| OP+ | - | - | - | 25.72 |
| OmniQuant | 37.50% | **35.60%** | **26.03%** | **26.93** |

Table A17: Performance of weights and activations quantization on LLaMA-1-7B model with asymmetric bits.

| #Bits | Method | PPL ↓ | | | Accuracy (%) ↑ | | | | | | |
|---|---|---|---|---|---|---|---|---|---|---|---|
| | | WikiText2 | C4 | Avg. | PIQA | ARC-e | ARC-c | BoolQ | HellaSwag | Winogrande | Avg. |
| FP16 | - | 5.68 | 7.08 | 6.38 | 77.47 | 52.48 | 41.46 | 73.08 | 73.00 | 67.07 | 64.09 |
| W4A8 | OmniQuant | 5.87 | 7.34 | 6.60 | 77.36 | 51.85 | 38.65 | 70.67 | 71.20 | 64.71 | 62.40 |
| W4A6 | OmniQuant | 6.09 | 7.63 | 6.85 | 75.73 | 51.51 | 38.31 | 68.28 | 70.79 | 65.27 | 61.64 |
| W8A4 | OmniQuant | 10.27 | 12.77 | 11.52 | 69.47 | 45.87 | 32.84 | 59.08 | 58.66 | 54.85 | 53.46 |

Table A18: **Weight-only quantization on Falcon-180B.**

| **Falcon-180b** | | | | PPL↓ | | | Acc↑ | | | | | |
|---|---|---|---|---|---|---|---|---|---|---|---|---|
| Method | Bit# | Memory | Devices | Wiki | PTB | C4 | PIQA | ARC-e | Arc-c | BoolQ | HellaSwag | Winogrande |
| - | BF16/FP16 | 335GB | 5xA100 80GB | 3.29 | 6.64 | 6.31 | 84.82 | 84.20 | 60.83 | 86.85 | 85.91 | 80.58 |
| RTN | W3A16g512 | 65GB | 1xA100 80GB | 5.33 | 8.08 | 8.34 | 83.48 | 80.85 | 55.46 | 78.37 | 81.05 | 77.97 |
| OmniQuant | W3A16g512 | **65GB** | **1xA100 80GB** | **3.71** | **6.95** | **6.71** | **84.71** | **82.91** | **60.92** | **84.03** | **84.96** | **79.40** |

Table A19: **C4 perplexity of Weight-only quantization results in LLaMA-1 and LLaMA-2 models** Continue of Table 1.

| **LLaMA1&2 / PPL↓** | | 1-7B | 1-13B | 1-30B | 1-65B | 2-7B | 2-13B | 2-70B |
|---|---|---|---|---|---|---|---|---|
| FP16 | - | 7.08 | 6.61 | 5.98 | 5.62 | 6.97 | 6.46 | 5.52 |
| | RTN | 1.3e5 | 5.6e4 | 2.7e4 | 2.2e4 | 4.8e4 | 7.2e4 | 2.4e4 |
| W2A16 | GPTQ | 689.13 | 2.5e3 | 169.80 | 40.58 | NAN | 323.12 | 48.82 |
| | **OmniQuant** | **24.89** | **18.31** | **13.89** | **10.77** | **90.64** | **26.76** | **12.28** |
| | RTN | 1.0e3 | 447.64 | 99.45 | 17.15 | 4.9e3 | 139.65 | 42.13 |
| W2A16 | GPTQ | 27.71 | 15.29 | 11.93 | 11.99 | 33.70 | 20.97 | NAN |
| g128 | AWQ | 1.9e5 | 2.3e5 | 2.4e5 | 7.5e4 | 1.7e5 | 9.4e4 | - |
| | **OmniQuant** | **12.97** | **10.36** | **9.36** | **8.00** | **15.02** | **11.05** | **8.52** |
| | RTN | 151.43 | 76.00 | 30.07 | 11.34 | 475.35 | 28.69 | 13.43 |
| W2A16 | GPTQ | 17.71 | 11.70 | 9.92 | 10.07 | 19.40 | 12.48 | NAN |
| g64 | AWQ | 2.8e5 | 2.2e5 | 2.3e5 | 7.4e4 | 1.6e5 | 9.5e4 | - |
| | **OmniQuant** | **11.78** | **9.75** | **8.65** | **7.60** | **12.72** | **10.05** | **7.88** |
| | RTN | 28.26 | 13.22 | 28.66 | 12.79 | 402.35 | 12.51 | 10.02 |
| W3A16 | GPTQ | 9.49 | 8.16 | 7.29 | 6.71 | 9.81 | 8.02 | 6.57 |
| | AWQ | 13.26 | 9.13 | 12.67 | 7.11 | 23.85 | 13.07 | - |
| | **OmniQuant** | **8.19** | **7.32** | **6.57** | **6.07** | **8.65** | **7.44** | **6.06** |
| | RTN | 8.62 | 7.49 | 6.58 | 6.10 | 8.40 | 7.18 | 6.02 |
| W3A16 | GPTQ | 7.85 | 7.10 | 6.47 | 6.00 | 7.89 | 7.00 | 5.85 |
| g128 | AWQ | 7.92 | 7.07 | 6.37 | 5.94 | 7.84 | **6.94** | - |
| | **OmniQuant** | **7.75** | **7.05** | **6.37** | **5.93** | **7.75** | 6.98 | **5.85** |
| | RTN | 7.93 | 6.98 | 6.34 | 5.85 | 7.71 | 6.83 | 5.79 |
| W4A16 | GPTQ | 7.43 | 6.84 | 6.20 | 5.80 | 7.37 | 6.70 | 5.67 |
| | AWQ | 7.52 | 6.86 | 6.17 | 5.77 | 7.68 | 6.74 | - |
| | **OmniQuant** | **7.34** | **6.76** | **6.11** | **5.73** | **7.35** | **6.65** | **5.65** |
| | RTN | 7.37 | 6.69 | 6.06 | 5.69 | 7.24 | 6.58 | 5.63 |
| W4A16 | GPTQ | 7.21 | 6.69 | 6.06 | 5.69 | 7.12 | 6.56 | 5.58 |
| g128 | AWQ | 7.21 | 6.70 | 6.05 | 5.68 | 7.13 | 6.56 | - |
| | **OmniQuant** | **7.21** | **6.69** | 6.06 | **5.68** | **7.12** | **6.56** | **5.58** |

Table A20: **WikiText2 perplexity of Weight-only quantization results in OPT models**.

| OPT / PPL↓ | | 125M | 1.3B | 2.7B | 6.7B | 13B | 30B | 66B |
|---|---|---|---|---|---|---|---|---|
| FP16 | - | 27.65 | 14.63 | 12.47 | 10.86 | 10.12 | 9.56 | 9.34 |
| W2A16 g128 | RTN | 7.2e3 | 1.3e4 | 5.7e4 | 7.8e3 | 7.6e4 | 1.3e4 | 3.6e5 |
| | GPTQ | 597.66 | 115.16 | 61.59 | 20.18 | 21.36 | 12.71 | 82.10 |
| | AWQ | 251.84 | 47.97 | 28.50 | 16.20 | 14.32 | 12.31 | 14.54 |
| | **OmniQuant** | **75.43** | **23.95** | **18.13** | **14.43** | **12.94** | **11.39** | 30.84 |
| W2A16 g64 | RTN | 7.0e3 | 1.0e4 | 19.3e4 | 7.6e3 | 1.8e4 | 8.2e3 | 1.1e4 |
| | GPTQ | 204.40 | 49.58 | 29.37 | 16.81 | 16.65 | 11.87 | 356.01 |
| | AWQ | 124.18 | 29.78 | 20.64 | 14.63 | 13.28 | 11.59 | 12.74 |
| | **OmniQuant** | **62.56** | **21.40** | **16.76** | **13.57** | **12.33** | **11.00** | **10.59** |
| W3A16 | RTN | 1.2e3 | 1.3e4 | 1.6e4 | 6.5e3 | 4.6e3 | 1.5e3 | 6.1 e3 |
| | GPTQ | 53.05 | 21.17 | 16.83 | 15.09 | 11.73 | 10.30 | 14.42 |
| | AWQ | 69.43 | 28.01 | 263.10 | 15.13 | 20.09 | 35.74 | 4.5e3 |
| | **OmniQuant** | **35.66** | **16.68** | **13.80** | **11.65** | **10.87** | **10.00** | **9.83** |
| W3A16 g128 | RTN | 51.22 | 119.00 | 297.98 | 23.54 | 46.03 | 18.80 | 136.89w |
| | GPTQ | 39.24 | 16.47 | 13.69 | 11.65 | 10.35 | 9.73 | 10.96 |
| | AWQ | 36.74 | 16.32 | 13.58 | 11.41 | 10.68 | 9.85 | 9.60 |
| | **OmniQuant** | **32.25** | **15.72** | **13.18** | **11.27** | **10.47** | **9.79** | **9.53** |
| W4A16 | RTN | 37.28 | 48.17 | 16.92 | 12.10 | 11.32 | 10.97 | 110 |
| | GPTQ | 31.43 | 15.56 | 12.82 | 11.41 | 10.31 | 9.63 | 9.55 |
| | AWQ | 32.28 | 15.49 | 12.93 | 11.30 | 10.39 | 9.77 | 9.61 |
| | **OmniQuant** | **29.45** | **15.04** | **12.76** | **11.03** | **10.30** | 9.65 | 9.65 |
| W4A16 g128 | RTN | 30.47 | 15.29 | 13.02 | 11.15 | 10.30 | 9.94 | 9.65 |
| | GPTQ | 29.81 | 14.89 | 12.52 | 10.93 | 10.17 | 9.58 | 9.34 |
| | AWQ | 29.15 | 14.94 | 12.74 | 10.93 | 10.21 | 9.59 | 9.40 |
| | **OmniQuant** | **28.86** | **14.88** | 12.65 | **10.96** | 10.20 | 9.62 | 9.37 |

Table A21: **PTB perplexity of Weight-only quantization results in OPT models**.

| OPT / PPL↓ | | 125M | 1.3B | 2.7B | 6.7B | 13B | 30B | 66B |
|---|---|---|---|---|---|---|---|---|
| FP16 | - | 32.54 | 16.96 | 15.11 | 13.08 | 12.33 | 11.84 | 11.36 |
| W2A16 g128 | RTN | 4.6e3 | 7.1e3 | 2.5e4 | 5.7e3 | 3.0e4 | 6.2e3 | 1.4e5 |
| | GPTQ | 655.17 | 130.88 | 61.36 | 25.24 | 20.46 | 15.15 | 323.23 |
| | AWQ | 263.88 | 71.87 | 43.15 | 19.49 | 17.61 | 14.92 | 19.33 |
| | **OmniQuant** | **126.49** | **34.33** | **25.28** | **18.92** | **16.74** | **14.51** | **139.17** |
| W2A16 g64 | RTN | 5.1e3 | 9.4e3 | 7.7e4 | 6.1e3 | 8.2e3 | 4.1e3 | 6.2e3 |
| | GPTQ | 245.28 | 55.61 | 36.12 | 19.45 | 17.02 | 14.05 | 88.92 |
| | AWQ | 143.18 | 41.19 | 25.08 | 18.00 | 15.83 | 14.92 | 15.72 |
| | **OmniQuant** | **112.10** | **30.36** | **22.63** | **17.58** | **15.70** | **13.98** | **13.51** |
| W3A16 | RTN | 1.2e3 | 1.1e4 | 1.0e4 | 5.2e3 | 3.6e3 | 1.4e3 | 3.6e3 |
| | GPTQ | 34.05 | 27.39 | 15.94 | 13.75 | 13.71 | 12.54 | 21.16 |
| | AWQ | 80.73 | 33.20 | 224.11 | 18.46 | 35.45 | 66.68 | 3.4e3 |
| | **OmniQuant** | **45.29** | **20.42** | **17.08** | **14.23** | **13.49** | **12.54** | **12.06** |
| W3A16 g128 | RTN | 64.67 | 222.13 | 337.75 | 39.90 | 65.33 | 34.27 | 309.69 |
| | GPTQ | 45.17 | 19.90 | 17.06 | 14.24 | 12.84 | 12.54 | 13.27 |
| | AWQ | 44.07 | 19.59 | 16.52 | 13.98 | 12.87 | 66.68 | 3.4e3 |
| | **OmniQuant** | **40.76** | **19.06** | **16.29** | **13.77** | **12.96** | **12.19** | **11.71** |
| W4A16 | RTN | 44.98 | 33.63 | 22.23 | 16.05 | 15.40 | 14.17 | 274.23 |
| | GPTQ | 37.75 | 18.23 | 15.94 | 13.75 | 12.58 | 11.98 | 11.58 |
| | AWQ | 38.74 | 18.35 | 15.70 | 13.59 | 12.72 | 12.06 | 11.58 |
| | **OmniQuant** | **34.94** | **17.80** | **15.52** | **13.41** | **12.62** | **11.95** | **11.86** |
| W4A16 g128 | RTN | 36.50 | 33.63 | 22.23 | 16.05 | 15.40 | 14.17 | 11.79 |
| | GPTQ | 35.48 | 17.41 | 15.42 | 13.21 | 12.42 | 11.89 | 11.51 |
| | AWQ | 34.95 | 17.46 | 15.33 | 13.28 | 12.46 | 11.90 | 11.43 |
| | **OmniQuant** | **34.28** | **17.40** | **15.28** | **13.25** | **12.46** | **11.94** | **11.40** |

Table A22: **C4 perplexity of Weight-only quantization results in OPT models**.

| OPT / PPL↓ | | 125M | 1.3B | 2.7B | 6.7B | 13B | 30B | 66B |
|---|---|---|---|---|---|---|---|---|
| FP16 | - | 24.60 | 14.72 | 13.16 | 11.74 | 11.19 | 10.69 | 10.28 |
| W2A16 g128 | RTN | 5.0e3 | 7.7e3 | 3.8e4 | 5.2e3 | 2.8e4 | 6.5e3 | 2.6e5 |
| | GPTQ | 597.66 | 60.88 | 33.83 | 18.55 | 16.34 | 12.89 | 598.81 |
| | AWQ | 168.35 | 38.38 | 26.41 | 16.48 | 14.73 | 12.98 | 15.42 |
| | **OmniQuant** | **80.10** | **27.33** | **21.11** | **16.67** | **14.92** | **13.12** | **73.83** |
| W2A16 g64 | RTN | 3.9e3 | 7.3e3 | 1.2e5 | 6.3e3 | 7.5e3 | 4.0e3 | 8.4e3 |
| | GPTQ | 133.51 | 31.31 | 23.23 | 16.24 | 14.48 | 12.24 | 58.60 |
| | AWQ | 90.19 | 27.34 | 20.01 | 15.20 | 13.90 | 12.43 | 13.31 |
| | **OmniQuant** | **64.01** | **23.71** | **19.16** | **15.44** | **14.16** | **12.80** | **12.13** |
| W3A16 | RTN | 722.83 | 6.1e3 | 1.2e4 | 5.8e3 | 3.3e3 | 1.4e3 | 3.6e3 |
| | GPTQ | 37.75 | 19.45 | 13.75 | 15.67 | 12.28 | 11.34 | 13.68 |
| | AWQ | 55.73 | 24.56 | 154.49 | 15.84 | 23.71 | 55.01 | 3.8e3 |
| | **OmniQuant** | **32.17** | **17.10** | **14.93** | **12.78** | **12.13** | **11.37** | **10.82** |
| W3A16 g128 | RTN | 40.13 | 126.47 | 372.23 | 32.56 | 44.12 | 25.70 | 286.87 |
| | GPTQ | 30.08 | 16.47 | 14.54 | 12.48 | 11.58 | 10.91 | 11.35 |
| | AWQ | 30.39 | 16.27 | 14.19 | 12.30 | 11.61 | 10.96 | 10.53 |
| | **OmniQuant** | **29.34** | **16.11** | **14.15** | **12.31** | **11.63** | **10.98** | **10.51** |
| W4A16 | RTN | 31.58 | 24.68 | 17.61 | 13.38 | 12.35 | 11.90 | 249.54 |
| | GPTQ | 27.12 | 15.57 | 13.75 | 12.15 | 11.36 | 10.80 | 10.50 |
| | AWQ | 27.64 | 15.65 | 13.71 | 12.04 | 11.42 | 10.83 | 10.41 |
| | **OmniQuant** | **26.36** | **15.28** | **13.58** | **11.97** | **11.41** | **10.80** | **10.63** |
| W4A16 g128 | RTN | 26.79 | 15.71 | 13.79 | 12.31 | 11.51 | 10.94 | 10.54 |
| | GPTQ | 25.96 | 15.05 | 13.40 | 11.87 | 11.26 | 10.74 | 10.37 |
| | AWQ | 25.90 | 15.04 | 13.39 | 11.87 | 11.28 | 10.75 | 10.34 |
| | **OmniQuant** | **25.63** | **15.03** | **13.38** | **11.85** | **11.29** | **10.75** | **10.33** |

Table A23: **WikiText2 perplexity of weight-activation quantization results in LLaMA-1 and LLaMA-2 models** Continue of Table 2.

| LLaMA1&2 / PPL↓ | | 1-7B | 1-13B | 1-30B | 1-65B | 2-7B | 2-13B |
|---|---|---|---|---|---|---|---|
| FP16 | - | 5.68 | 5.09 | 4.10 | 3.53 | 5.47 | 4.88 |
| W6A6 | SmoothQuant | 6.03 | 5.42 | 4.55 | 3.88 | 6.20 | 5.18 |
| | **OmniQuant** | **5.96** | **5.28** | **4.38** | **3.75** | **5.87** | **5.14** |
| W4A4 | SmoothQuant | 25.25 | 40.05 | 192.40 | 275.53 | 83.12 | 35.88 |
| | **OmniQuant** | **11.26** | **10.87** | **10.33** | **9.17** | **14.26** | **12.30** |

Table A24: **C4 perplexity of weight-activation quantization results in LLaMA-1 and LLaMA-2 models**. Continue of Table 2.

| LLaMA1&2 / PPL↓ | | 1-7B | 1-13B | 1-30B | 1-65B | 2-7B | 2-13B |
|---|---|---|---|---|---|---|---|
| FP16 | - | 7.08 | 6.61 | 5.98 | 5.62 | 6.97 | 6.46 |
| W6A6 | SmoothQuant | 7.47 | 6.97 | 6.34 | 5.99 | 7.76 | 6,76 |
| | **OmniQuant** | **7.43** | **6.84** | **6.22** | **5.82** | **7.48** | **6.74** |
| W4A4 | SmoothQuant | 32.32 | 47.18 | 122.38 | 244.35 | 77.27 | 43.19 |
| | **OmniQuant** | **14.51** | **13.78** | **12.49** | **11.28** | **18.02** | **14.55** |

Table A25: **Weight-activation quantization results of OPT Models.** We report perplexity on three datasets: WikiText2 (WIKI), Pen Treebank (PT), and C4. RPTQ indicates the data from RPTQ (Yuan et al. (2023)) paper, which keeps the output of LN and SoftMax as 8-bit. RPTQ* represents reproducing RPTQ with our setting that quantizes all activation into low-bit except keeping the softmax output at full precision.

| OPT / PPL↓ | | OPT-6.7b | | | OPT-13b | | | OPT-30b | | | OPT-66b | | |
|---|---|---|---|---|---|---|---|---|---|---|---|---|---|
| Task | | WIKI | PT | C4 | WIKI | PT | C4 | WIKI | PT | C4 | WIKI | PT | C4 |
| FP16 | - | 10.86 | 13.09 | 11.74 | 10.13 | 12.34 | 11.20 | 9.56 | 11.84 | 10.69 | 9.34 | 11.36 | 10.28 |
| W6A6 | SmoothQuant | 11.34 | 13.82 | 12.14 | 10.56 | 12.76 | 11.40 | 9.67 | 12.01 | 10.81 | 10.72 | 13.25 | 11.60 |
| | RPTQ | 11.19 | 13.98 | 12.08 | 11.00 | 15.23 | 11.68 | 10.22 | 14.95 | 11.73 | 9.45 | 13.03 | 10.62 |
| | RPTQ* | 10.96 | 13.24 | 11.86 | 10.25 | 12.60 | 11.31 | 9.60 | 12.23 | 10.83 | 9.48 | 12.61 | 10.39 |
| | **OmniQuant** | **10.96** | **13.20** | **11.81** | **10.21** | **12.47** | **11.27** | **9.62** | **11.92** | **10.76** | **9.42** | **11.42** | **10.32** |
| W4A4 | SmoothQuant | 1.8e4 | 1.4e4 | 1.5e4 | 7.4e3 | 6.5e3 | 5.6e3 | 1.2e4 | 7.8e3 | 8.3e3 | 2.2e5 | 1.0e5 | 1.8e5 |
| | RPTQ | 12.00 | 15.17 | 12.85 | 12.74 | 15.76 | 14.71 | 11.15 | 14.11 | 13.48 | 12.23 | 18.87 | 15.93 |
| | RPTQ* | 17.83 | 25.10 | 19.91 | 16.45 | 23.01 | 16.80 | 11.50 | 14.87 | 12.81 | 11.16 | 13.73 | 11.78 |
| | **OmniQuant** | **12.24** | **15.54** | 13.56 | **11.65** | **15.89** | 13.46 | **10.60** | **13.75** | **11.89** | **10.29** | **13.19** | **11.35** |

