# OpenReview forum: "OmniQuant: Omnidirectionally Calibrated Quantization for Large Language Models"
_ICLR.cc/2024/Conference — ICLR 2024 spotlight_

### Official Review · Reviewer_PTbL · 2023-10-30

**Soundness:** 3 good
**Presentation:** 3 good
**Contribution:** 3 good
**Rating:** 6
**Confidence:** 5

**Summary:**

The paper introduces "OmniQuant," a quantization method of Large Language Models (LLMs) for efficient deployment. Unlike traditional post-training quantization (PTQ) methods that manually select quantization parameters, OmniQuant learns these parameters, enabling effective low-bit quantization. It features two main components: Learnable Weight Clipping (LWC) which adjusts the clipping thresholds, and Learnable Equivalent Transformation (LET) that shifts quantization challenges from activations to weights. OmniQuant operates within a differentiable framework, making it efficient for both weight-only and weight-activation quantization. Experiments on OPT, LLaMA-1, LLaMA-2, and Falcon model family demonstrate the effectiveness of the proposed method.

**Strengths:**

The paper is well-written and easy to follow. The proposed method, though simple in design, proves to be remarkably effective, notably diminishing the performance degradation for low-bitwidth quantization. The proposed OmniQuant does not introduce extra computation or parameters for the quantized model since the introduced learnable parameters can be fused into quantized weights.

**Weaknesses:**

The major weakness of the paper is the noticeable absence of experimental comparisons with Outlier Suppression+ (OS+) (Wei et al., 2023). Despite OmniQuant's learnable equivalent transformation being conceptually similar to OS+, the paper does not provide a direct comparison or a detailed discussion highlighting the distinctions between the two methods. Such a comparison would be invaluable for readers and would significantly augment the paper's credibility and depth.

**Questions:**

1.	The novelty of the proposed learnable equivalent transformation is limited as the main idea learning channel-wise shifting and scaling is similar to Outlier Suppression+ (OS+) (Wei et al., 2023). A comparative discussion elucidating the distinctions between OmniQuant and OS+ would be beneficial for readers. Additionally, the absence of experimental comparisons with OS+ is a notable omission that should be addressed.

2.	The proposed LWC learns a clipping strength instead of clipping threshold in PACT (Choi et al., 2018) and LSQ (Esser et al., 2019). However, the paper lacks a clear articulation of the advantages of LWC over PACT and LSQ, particularly in scenarios where it is combined with LET and weights are frequently changed. A more thorough explanation of the benefits and underlying mechanics of LWC in such contexts would be beneficial. Additionally, an investigation into whether an iterative application of LWC and LET would yield performance improvements could provide valuable insights.

3.	In Section 3.1, the authors delineate the incorporation of learnable parameters, denoted as $\gamma$ and $\beta$, to learn the clipping threshold. While the methodology is clear, the experimental section does not furnish a thorough illustration of these parameters' distribution across layers. An inclusion of this visualization would strengthen the paper.

4.	The authors apply the LET to all linear layers, with the notable exception of the second linear layer of the FFN within the proposed method. This selective application raises an intriguing question: Do all instances of LET actively contribute to the model's final performance? An investigation into the individual and cumulative impact of LET on each linear layer could provide deeper insights into the efficacy and necessity of LET across different layers of the model.

5.	In the experiments, the authors mention retaining the Softmax output at full-precision owing to its long-tail distribution. It would be insightful to know the implications of quantizing the Softmax output to 8-bit. How does this quantization impact the overall model performance and accuracy?

6.	In the experimental section, the authors mention initializing the channel-wise scaling factor using SmoothQuant (Xiao et al., 2023) and the channel-wise shifting factor with Outlier Suppression+. A pertinent question arises: Is the proposed method sensitive to these initializations? It would be elucidative to explore the effects when the channel-wise scaling factor is initialized to 1 and the channel-wise shifting factor to 0. How does this affect the quantization performance?

---

> ### Author Response · Authors · 2023-11-17
> **Response to Reviewer PTbL (Part: 1/3)**
>
> We sincerely appreciate the time and efforts you have dedicated to reviewing our paper. Should you have any further inquiries, please let us know and we are more than delighted to discuss with you and run more experiments for any pieces of your interests in our work.Reviewer PTbL
>
> We appreciate your valuable insights and suggestions. We have summarized the updating of our revision in [Summary of Paper Updating](https://openreview.net/forum?id=8Wuvhh0LYW&noteId=YV1oX701CV) and provided an in-depth discussion about the novelty of our method in [Genera Response](https://openreview.net/forum?id=8Wuvhh0LYW&noteId=D6esYYnEi2). Additionally, we have made the following improvements to address your concerns.
>
> **Q1:** The major weakness of the paper is the noticeable absence of experimental comparisons with Outlier Suppression+ (OS+) (Wei et al., 2023).
>
> **A1:** We have added the comparisons with OS+ in Table 2 of our manuscript. A comparison overview is as follows.
>
> | W4A4        | Quantization | Average Acc. |
> | ----------- | ------------ | ------------ |
> | LLaMa-1-7B  | OS+          | 48.43        |
> | LLaMa-1-7B  | OmniQuant    | 52.65        |
> | LLaMa-1-13B | OS+          | 49.86        |
> | LLaMa-1-13B | OmniQuant    | 54.37        |
> | LLaMa-1-30B | OS+          | 52.62        |
> | LLaMa-1-30B | OmniQuant    | 56.63        |
> | LLaMa-1-65B | OS+          | 52.52        |
> | LLaMa-1-65B | OmniQuant    | 59.23        |
>
>
>
>
>
> **Q2:** Distinctions with Outlier Supression+.
>
> **A2:** OmniQuant differs from Outlier Suppression+ in several aspects, including parameter optimization through gradient updates, block-wise optimization, expansion of equivalent transformation to attention operations, and the introduction of learnable weight clipping (LWC). Details are as follows.
>
> - Both OmniQuant and OS+ try to find the optimal equivalent transformation parameters. OS+ solves this problem by pre-defined shifting strength and grid-searched scaling parameters. However, OmniQuant obtains both shifting and scaling parameters through gradient optimization.
>
> - OmniQuant leverages block-wise optimization while OS+ executes grid searching through a mix of single linear layer and multiple linear layers objective(Eq.6 and Eq.7 in their paper). Block-wise optimization considers interaction within a block and can produce better performance as demonstrated in BRECQ. However, it is a challenge for OS+ to expand to block-wise quantization, which would significantly enlarge the solution space and increase the time of grid searching.
>
> - LET expends the equivalent transformation to attention operation (Eq. (5) in our paper), but OS+ only does the equivalent transformation on linear layers.
>
> - OmniQuant also introduces learnable weight clipping (LWC), which further facilitates the low-bits (2,3) weight-only quantization, while Outlier Suppression only supports weight-activation quantization.
>
> We have added this discussion in Section A2 in our revision. You can also refer [Genera Response]() for more discussion about the novelty of our paper.
>
>
>
> **Q3:** The proposed LWC learns a clipping strength instead of clipping threshold in PACT (Choi et al., 2018) and LSQ (Esser et al., 2019). However, the paper lacks a clear articulation of the advantages of LWC over PACT and LSQ, particularly in scenarios where it is combined with LET and weights are frequently changed. A more thorough explanation of the benefits and underlying mechanics of LWC in such contexts would be beneficial.
>
> **A3:** To clarify the advantages of LWC, we have included Figure A5 in the appendix, illustrating the dynamic change in weight ranges during training with different clipping-based methods. The figure emphasizes that LET, by dramatically altering weight ranges, poses a challenge for traditional absolute clipping thresholds or step size learning. In contrast, LWC calculates relative clipping strength, utilizing extremums after LET as proxies. This approach enables LWC to adapt to changing weight ranges more naturally, addressing the optimization of difficulty introduced by LET.

---

> ### Author Response · Authors · 2023-11-17
> **Response to Reviewer PTbL (Part: 2/3)**
>
> **Q4:** Additionally, an investigation into whether an iterative application of LWC and LET would yield performance improvements could provide valuable insights.
>
> **A4:** We want to emphasize that in our approach, LWC and LET are trained simultaneously, and we have also explored an iterative training approach by iterations or epochs. The results, as presented in the following table, clearly indicate that training LWC and LET simultaneously yields the best performance. As aforementioned in Q3, LWC leverages the extremums after LET as proxies, allowing it to capture the dynamic changes in the weight range introduced by LET. This synergy between LET and LWC creates a progressive process, where both techniques reinforce each other rather than interfere. To further support this statement, we conducted an additional experiment (last row in the table), training LWC and LET iteratively with double the epochs. The results show that simultaneous training with 20 epochs achieves comparable performance to iterative training with 40 epochs. This demonstrates the effectiveness and efficiency of training LWC and LET simultaneously. We have added this experiment in Table A8 of our revision.
>
> | LLaMa-1-7B/ W4A4/ Iterative LWC and LET | Average PPL | Average Acc. |
> | --------------------------------------- | ----------- | ------------ |
> | simultaneous training (OmniQuant)       | 12.87       | 52.65        |
> | each iteration                          | 13.56       | 50.91        |
> | each epoch                              | 13.51       | 52.06        |
> | each epoch + double training epochs     | 12.80       | 52.50        |
>
>
>
> **Q5**: Visualization of $\gamma$ and $\beta$ distribution.
>
> **A5:** Indeed, we have plotted the distribution of learned \gamma and \beta in Figure A1 and presented the analysis in Section A5 in the Appendix. Through the visualization in Figure A1, we can find that LWC can learn more pronounced clipping when meeting with lower quantization precision (lower bits or greater group size). Such a phenomenon makes sense due to that lower quantization precision has smaller representation space, and consequently is expected to clip more weights to obtain better trade-offs.
>
>
>
> **Q6:** An investigation into the individual and cumulative impact of LET on each linear layer could provide deeper insights into the efficacy and necessity of LET across different layers of the model.
>
> **A6:** Good question. In our paper, we exclude the LET of the second linear layer due to the high sparsity of features after the non-linear layer leads to unstable gradients. Therefore, we have four LET pairs, represented as [ln1, (q_proj, k_proj, v_proj)], [v_proj, out_proj], [Q, K], and [ln2, fc1]. To validate the effect of each part, we respectively remove the LET from each layer and report the average perplexity and the average accuracy on 6 zero-shot tasks like Table 2. As shown in the following table, we can find that all four LETs can improve the performance, specially for the [ln1, (q_proj, k_proj, v_proj)] pair. Such results also demonstrate that the activation outliers are more serious after layer normalization layers. We have added this experiment in Table A5 of our revision.
>
> | LLaMa-1-7B                              | Average PPL | Average Acc. |
> | --------------------------------------- | ----------- | ------------ |
> | FP                                      | 6.38        | 64.09        |
> | W4A4 (ours)                             | 12.87       | 52.65        |
> | W4A4  - [ln1, (q_proj, k_proj, v_proj)] | 19.87       | 46.79        |
> | W4A4  -[v_proj, out_proj]               | 13.03       | 51.68        |
> | W4A4  -[Q, K]                           | 13.34       | 51.47        |
> | W4A4  -[ln2, fc1]                       | 14.47       | 51.04        |

---

> ### Author Response · Authors · 2023-11-17
> **Response to Reviewer PTbL (Part: 3/3)**
>
> **Q7:** In the experiments, the authors mention retaining the Softmax output at full-precision owing to its long-tail distribution. It would be insightful to know the implications of quantizing the Softmax output to 8-bit. How does this quantization impact the overall model performance and accuracy?
>
> **A7**: We have conducted experiments on quantizing the Softmax output, reporting the impact on model performance. As shown in the following table, we can find that quantizing the softmax into 8-bit and 6-bit brings acceptable performance degeneration, which demonstrates that block-wise calibration can compensate for the loss of 8-bit and 6-bit Softmax quantization. However, 4-bit Softmax quantization brings significant performance loss, which requires further exploration and additional trick such as log2 quantization in RepQViT[1]. We have added this experiment in Table A7 of our revision.
>
> | LLaMa-1-7B           | Average PPL | Average Acc. |
> | -------------------- | ----------- | ------------ |
> | FP                   | 6.38        | 64.09        |
> | W4A4 + Softmax 16bit | 12.87       | 52.65        |
> | W4A4 + Softmax 8bit  | 12.91       | 51.93        |
> | W4A4 + Softmax 6bit  | 13.20       | 51.70        |
> | W4A4 + Softmax 4bit  | 18.80       | 48.52        |
>
>
>
> **Q8:**  Sensitivity to channel-wise scaling and shifting initialization.
>
> **A8**: Good advice, sorry for the missing of this ablation. As shown in the following table, we can find that both careful initialization of scaling and shifting can improve the final performance. Specifically, scaling initialization is more important than shifting due to scaling plays the main role in alleviating outliers. We have added this experiment in Table A6 of our revision.
>
> | LLaMa-1-7B W4A4          | Average PPL | Average Acc. |
> | ------------------------ | ----------- | ------------ |
> | careful initialization   | 12.87       | 52.65        |
> | initialize scaling as 1  | 13.64       | 51.37        |
> | initialize shifting as 0 | 12.95       | 52.22        |
>
>
>
> We sincerely appreciate the time and efforts you have dedicated to reviewing our paper. Should you have any further inquiries, please let us know and we are more than delighted to discuss with you and run more experiments for any pieces of your interests in our work.
>
>
>
> [1] Repq-vit: Scale reparameterization for post-training quantization of vision transformers

---

> ### Author Response · Authors · 2023-11-20
> **Look forward to further discussion.**
>
> Dear Reviewer PTbL,
>
> We hope that our response can address your concerns. As the deadline for discussion period is approaching, we really appreciate if you can let us know whether there still exists any further question about the paper or the response. We are looking forward to further discussion.
>
> Best regards,
>
> Authors of Paper 1585

---

> > ### Comment · Reviewer_PTbL · 2023-11-20
> > **Response to rebuttal**
> >
> > Thanks to the authors for their informative rebuttal, which has successfully addressed my concerns. In light of this, I am inclined to increase my rating to 6.

---

> > > ### Author Response · Authors · 2023-11-20
> > > **Thanks for your feedback.**
> > >
> > > Dear Reviewer PTbL,
> > >
> > > Thank you for your feedback! We are pleased to address your concerns and greatly appreciate your reviews, which play a crucial role in improving our work.
> > >
> > > Best regards,
> > >
> > > Authors of Paper 1585

---

### Official Review · Reviewer_ngyH · 2023-10-30

**Soundness:** 3 good
**Presentation:** 2 fair
**Contribution:** 3 good
**Rating:** 8
**Confidence:** 5

**Summary:**

OmniQuant represents Learnable Weight Clipping (LWC) and Learnable Equivalent Transformation (LET). In contrast to the traditional min-max scaling threshold, LWC applies a sigmoid function on the factors over min and max. The loss is computed by comparing the distortion of the output, which can be easily back-propagated to the scaling factors. Similar work on weight clipping could be found in PACT [1] and LSQ [2]. However, OmniQuant's formulation is cleaner (no need to incorporate with learnable step-size) and more general (applies to both weight and activation).

The LET in OmniQuant decides the computation operands ordering for max hardware efficiency. The implementation is based on MLC [3].

* [1] Choi, J., Wang, Z., Venkataramani, S., Chuang, P.I.J., Srinivasan, V. and Gopalakrishnan, K., 2018. Pact: Parameterized clipping activation for quantized neural networks. arXiv preprint arXiv:1805.06085.
* [2] Esser, S.K., McKinstry, J.L., Bablani, D., Appuswamy, R. and Modha, D.S., 2019. Learned step size quantization. arXiv preprint arXiv:1902.08153.
* [3] Feng, S., Hou, B., Jin, H., Lin, W., Shao, J., Lai, R., Ye, Z., Zheng, L., Yu, C.H., Yu, Y. and Chen, T., 2023, January. Tensorir: An abstraction for automatic tensorized program optimization. In Proceedings of the 28th ACM International Conference on Architectural Support for Programming Languages and Operating Systems, Volume 2 (pp. 804-817).

**Strengths:**

The biggest contribution of this work is the learnable weight clipping (LWC).

1. When compared to AWQ [1] which only optimize the scale, LWC uses the same objective function (minimize output distortion) but also handles clipping.

2. When compared to PACT, which only applies to activation with positive values, LWC is more general and applies to weight, activation, and negative values.

3. When compared to LSQ, which is a combination of multiple optimization goals (step-size, gradient scaling, and clipping), LWC is straightforward to implement and speedy to train.

[1] Lin, J., Tang, J., Tang, H., Yang, S., Dang, X. and Han, S., 2023. AWQ: Activation-aware Weight Quantization for LLM Compression and Acceleration. arXiv preprint arXiv:2306.00978.

**Weaknesses:**

1. While I highlight the advantage of LWC over AWQ, PACK, and LSQ, this evaluation for PACK and LSQ is lacking.

2. Authors should present Figure 1 (a) in a quantitative way. It lacks actual numbers for cost of time and performance.

3. Learnable Equivalent Transformation (LET) isn't novel and can be seen in common quantization kernels such as LLM.int8. I suggest authors to elaborate more on comparison with PACK and LSQ instead of LET.

4. The latency benchmark only contain fp16 and OmniQuant. Strong baseline such as GPTQ is lacking.

**Questions:**

1. How does OmniQuant runtime efficiency compare to GPTQ and AWQ?

2. What is the performance when compared to PACK and LSQ?

---

> ### Author Response · Authors · 2023-11-17
> **Response to Reviewer ngyH (Part: 1/2)**
>
> Thanks for your valuable and supportive reviews. We have summarized the updating of our revision in [Summary of Paper Updating](https://openreview.net/forum?id=8Wuvhh0LYW&noteId=YV1oX701CV) and provided an in-depth discussion about the novelty of our method in [Genera Response](https://openreview.net/forum?id=8Wuvhh0LYW&noteId=D6esYYnEi2). Below, we explain our method and contribution more clearly in Q1 to Q2 and address the weakness from Q3 to Q6 as follows.
>
> **Q1:** The LET in OmniQuant decides the computation operands ordering for max hardware efficiency. The implementation is based on MLC.
>
> **A1:** With full respect, Learnable Equivalent Transformation (LET) in OmniQuant focuses on channel-wise scaling and shifting to facilitate low-bits quantization by transferring quantization difficulty between weights and activation. Importantly, LET is independent of hardware efficiency considerations. After training, all LET and LWC parameters are absorbed into the original module without introducing additional memory or computational burden. Therefore, OmniQuant's quantized models can seamlessly deploy through existing frameworks such as MLC-LLM.
>
>
>
> **Q2:** The biggest contribution of this work is the learnable weight clipping (LWC).
>
> **A2:** With all due respect, the novelty and primary contribution of OmniQuant lies in being the first to introduce differentiable gradient optimization into learnable equivalent transformation (LET) and learnable weight clipping (LWC) for large language models. Previous PTQ methods for LLMs such as Outlier Suppression+ or AWQ solve such problems by grid searching, which limits the performance in low-bit scenarios. Furthermore, while the concepts of equivalent transformation and clipping are not new, we have made some innovative improvements to make the optimization process more stable. For equivalent transformation, we expend the equivalent transformation from linear layer to attention operation (Eq. (5) in our paper). For weight clipping, we propose taking the extremums after LET as proxies, which helps LWC to catch the dramatic changs led by LET as described in Section A5. These advancements enable OmniQuant to excel in both weight-only and weight-activation quantization scenarios.
>
>
>
> **Q3:** While I highlight the advantage of LWC over AWQ, PACT, and LSQ, this evaluation for PACT and LSQ is lacking. What is the performance when compared to PACK and LSQ?
>
> **A3:**   As described in Algorithm 1 in Appendix, OmniQuant firstly leverages LET to transfer magnitude from activation to weight, and then quantizes the weight after LET with LWC. To demonstrate the advantage of LWC, we replace LWC in our pipeline with PACT (train the clipping threshold directly) and LSQ (train the quantization step size directly) in Table A14, showcasing that LWC consistently outperforms PACT and LSQ. Additionally, Figure A5 in the Appendix further clarifies the advantages of LWC by illustrating its adaptability to changing weight ranges introduced by LET, addressing optimization difficulties not addressed by traditional methods. Please also see Q2/A2 in responses to WDuc.

---

> ### Author Response · Authors · 2023-11-17
> **Response to Reviewer ngyH (Part: 2/2)**
>
> **Q4:** Authors should present Figure 1 (a) in a quantitative way. It lacks actual numbers for cost of time and performance.
>
> **A4:** Thank you for the suggestion. We have incorporated quantitative results into Figure 1(a) to provide a more comprehensive understanding of the cost of time and performance.
>
>
>
> **Q5:**  Learnable Equivalent Transformation (LET) isn't novel and can be seen in common quantization kernels such as LLM.int8. I suggest authors to elaborate more on comparison with PACK and LSQ instead of LET.
>
> **A5:** As mentioned in Q2, LET is the first one to introduce gradient optimization into the equivalent transformation of large language models, which significantly pushes the limitation of low-bits quantization. To demonstrate the superiority of the proposed method, we have compared it with state-of-the-art weight-only (GPTQ, AWQ) and weight-activation (LLM-QAT, Outlier Suppression +) quantization methods in our paper. Additionally, we have transferred PACT and LSQ to the weight quantization in Table A14, showcasing that LWC outperforms previous clipping-based methods. As for the activation quantization, we leverage MinMax quantization due to that directly leverages LSQ in activation quantization leading to poor performance than MinMAx quantization as shown in Table 4 of the LLM-QAT paper. You can also refer [Genera Response](https://openreview.net/forum?id=8Wuvhh0LYW&noteId=D6esYYnEi2) for more discussion about the novelty of our paper.
>
>
>
>
>
> **Q6:** The latency benchmark only contain fp16 and OmniQuant. Strong baseline such as GPTQ is lacking. How does OmniQuant runtime efficiency compare to GPTQ and AWQ?
>
> **A6:** OmniQuant, GPTQ, and AWQ are all uniform quantization methods, allowing their quantized models to be deployed with the same latency. In our paper, the weight-only quantization kernel is not our primary focus, and we leverage existing MLC-LLM to validate the practical deployment of our proposed methods. As shown in the following table, we present the latency compared between MLC-LLM, GPTQ kernel, and AWQ kernel. Note that all of these three manners can be used to deploy OmniQuant’s quantized models.
>
> |                  | LLaMa-7B      | LLaMa-13B     |
> | ---------------- | ------------- | ------------- |
> | FP               | 69.2 token/s  | 52.5 token/s  |
> | 4bit-MLC-LLM     | 134.2 token/s | 91.3 token/s  |
> | 4bit-GPTQ kernel | 91.30 token/s | 63.2 token /s |
> | 4bit-AWQ kernel  | 155.4 token/s | 94.86 token/s |
>
>
>
> We sincerely appreciate the time and efforts you have dedicated to reviewing our paper. Should you have any further inquiries, please let us know and we are more than delighted to discuss with you and run more experiments for any pieces of your interests in our work.

---

> > ### Comment · Reviewer_ngyH · 2023-11-20
> >
> > I thank authors for the rebuttal. The updated benchmarks and appendix addressed my concern and I increased my rating from 6 to 8.

---

> > > ### Author Response · Authors · 2023-11-22
> > > **Thanks for your feedback.**
> > >
> > > Dear Reviewer ngyH,
> > >
> > > Thank you for your feedback! We are pleased to address your concerns and greatly appreciate your reviews, which play a crucial role in improving our work.
> > >
> > > Best regards,
> > >
> > > Authors of Paper 1585

---

### Official Review · Reviewer_WDuc · 2023-10-31

**Soundness:** 2 fair
**Presentation:** 3 good
**Contribution:** 2 fair
**Rating:** 6
**Confidence:** 4

**Summary:**

The paper works on quantization for large language models. It first proposes to learn the weight clipping threshold with optimization on the ratio of weight ranges. Then, it proposes to learn the equivalent parameters for learnable equivalent transformation with a block-wise loss. Experiments are done on 4-bit weight activation quantization and 4-bit weight-only quantization with LLaMA and OPT models. Especially, the paper evaluates the inference speed with 4-bit weight-only quantized models.

**Strengths:**

* Experiments are done across several datasets including common sense reasoning and perplexity evaluation. Also, the paper tries the hard setting with 4-bit weight and activation quantization. Especially, it evaluates the latency with a 4-bit weight quantized model.
* The structure of the paper is clear and figures are drawn well.
* The method is simple and considers the quantization difficulties both for weights and activations.

**Weaknesses:**

* The paper lacks a necessary detailed explanation for the motivation and effectiveness of the weight clipping method.
  * In 3.2, the paper claims that directly employing prior LSQ and PACT would produce unsatisfactory performance, as demonstrated in LLM-QAT. However, LLM-QAT says that the outliers for activation have a notable impact, bringing difficulty for clipping while this method works on weights here. Also, how can the proposed Eq. (2) solve the problem of learning clipping thresholds for outliers? In other words, what is the optimization difficulty (concept given in 3.2) of previous techniques, and how can Eq.(2) solve it? More explanation about the motivation is preferred.
  * In the appendix, the paper says that LET would decline the convergence of LSQ and PACT because LET alters the weight distribution. However, weight distribution altering is a common case for LSQ and PACT in QAT. Also, combined with LET, the \gamma and \beta in the proposed Eq. (2) can also go up and down during learning as it optimizes the ratio of the changeable weig

* What is the core novelty of the LET? I find it looks similar to Outlier Suppression+. While the paper says that Outlier Suppression+ takes a pre-defined migration strength, but this method does not and proposes to optimize the output. I'd like to point out that Outlier Suppression+ did not take a pre-defined strength and proposed to optimize the output for channel-wise scaling parameters earlier than this paper. Meanwhile, the paper also states that AWQ adopts a grid-searched channel-wise scaling, which also seems relevant to the technique in this paper. Therefore, can the paper compare these different designs and explain why the proposed way is the best? I did not find these and this could help us better understand the effectiveness.

* Experiments shall be compared with the paper Outlier Suppression+ because you and they work on the same quantization problem, take the same equivalent transformation, and have similar optimization designs.

* I noticed that the paper requires careful equivalent parameter initialization via the compared baseline SmoothQuant. I might wonder how it behaves without good initialization. For example, under asymmetric cases W4A8, W4A6, and W8A4, where the LLM-QAT shows SmoothQuant can behave terribly.

* To conclude, I find the proposed two techniques are not novel and the paper lacks the necessary explanation and comparison. I see the challenge of the two techniques is how to combine the two kinds of learning together as they influence each other. However, current techniques seem can not solve this problem well. Thus, I think it would be better if the paper gives more description, and design consideration to the combination part, which might increase the novelty. For example, maybe alternately train these two techniques.

**Questions:**

Please check the weakness part.

---

> ### Author Response · Authors · 2023-11-17
> **Response to Reviewer WDuc (Part: 1/3)**
>
> We would like to express our sincere gratitude to Reviewer WDuc for their valuable time and effort in reviewing our work. We have summarized the updating of our revision in [Summary of Paper Updating](https://openreview.net/forum?id=8Wuvhh0LYW&noteId=YV1oX701CV) and provided an in-depth discussion about the novelty of our method in [Genera Response](https://openreview.net/forum?id=8Wuvhh0LYW&noteId=D6esYYnEi2). Below, we address each identified weakness.
>
> **Q1:** In 3.2, the paper claims that directly employing prior LSQ and PACT would produce unsatisfactory performance, as demonstrated in LLM-QAT. However, LLM-QAT says that the outliers for activation have a notable impact, bringing difficulty for clipping while this method works on weights here.
>
> **A1:** We appreciate your clarification regarding LLM-QAT. Indeed, the confusion arises from our initial claim about the unsatisfactory performance of LSQ and PACT in weight quantization in Sec. 3.2 and Table A14. We have modified the expression to address any potential confusion. Thanks for the correction.
>
>
>
> **Q2:**  How can the proposed Eq. (2) solve the problem of learning clipping thresholds for outliers? In other words, what is the optimization difficulty (concept given in 3.2) of previous techniques, and how can Eq.(2) solve it? More explanation about the motivation is preferred. In the appendix, the paper says that LET would decline the convergence of LSQ and PACT because LET alters the weight distribution. However, weight distribution altering is a common case for LSQ and PACT in QAT. Also, combined with LET, the \gamma and \beta in the proposed Eq. (2) can also go up and down during learning as it optimizes the ratio of the changeable weight.
>
> **A2:** Thanks for your question. We claim that LWC reduces the optimization difficulty in learning clipping thresholds because our LWC reparameterizes the learnable  clipping threshold into the product of a learnable 0~1 clipping strength and the  maximum or minimum of weights. The benefit is that no matter how the maximum and minimum of weights change caused by LET with the training going, our LWC would always learn a clipping strength under the reference of the current maximum and minimum of weights . Such a reparameterization technique has been utilized in various areas for better training properties. For example, batch normalization decomposes the neural activation into a variable approximately following normal distribution and learnable scale and shift parameters, which improves the training stability.
>
> Here we provide more empirical evidence to clarify the motivation and advantages of LWC. As shown in Figure A5 in the appendix, it presents the dynamic change in weight ranges (maximun minus mininum) during training with different clipping-based methods. We can see that the weight range varies a lot as the training goes on (see the curve of Min-Max) when training our OmniQuant, which is caused by our proposed another component LET. It poses a challenge for traditional absolute clipping thresholds (PACT) or step size learning (LSQ). As we can see from Figure A5 (a & b), no matter how the weight range of Min-Max increases, PACT and LSR would always retain a relatively small weight range, which would clip many regular weights values to the threshold. In contrast, LWC calculates relative clipping strength and enables it to adapt to varying weight ranges, as shown in Figure A5 (c).
>
>  We hope that our explanations and visualizations will address your concerns.

---

> ### Author Response · Authors · 2023-11-17
> **Response to Reviewer WDuc (Part: 2/3)**
>
> **Q3:** What is the core novelty of the LET? I find it looks similar to Outlier Suppression+. While the paper says that Outlier Suppression+ takes a pre-defined migration strength, but this method does not and proposes to optimize the output. I'd like to point out that Outlier Suppression+ did not take a pre-defined strength and proposed to optimize the output for channel-wise scaling parameters earlier than this paper. Meanwhile, the paper also states that AWQ adopts a grid-searched channel-wise scaling, which also seems relevant to the technique in this paper. Therefore, can the paper compare these different designs and explain why the proposed way is the best? I did not find these and this could help us better understand the effectiveness.
>
> **A3:** We have provided a detailed comparison of LET with Outlier Suppression+ (OS+) and AWQ in the table presented below.
>
> - For the equivalent transformation, AWQ only considers scaling operation, while OS+ and our LET consider both scaling and shifting operation.
>
> - For the execution position, AWQ and OS+ only carry equivalent transformation on linear layers, while LET also considers the matrix multiplication within attention (Eq. (5) in our paper). This point enlarges the solution space of equivalent transformation and brings further improvements as demonstrated in Table A5 of Appendix.
>
> -  For the optimization, OS+ leverages pre-defined shifting, both AWQ and OS+ find the scaling factors through grid searching based on some heuristic strategy. However, LET optimized all equivalent transformation parameters through end-to-end gradient descent, which significantly improve the performance. As acknowledged by reviewer ZyAK, making the learning of equivalent transformation stable and effective is a good contribution. Additionally, OmniQuant leverages block-wise optimization objectives, while both AWQ and OS+ take layer-wise optimization objectives. Block-wise optimization considers interaction within a block and can produce better performance as demonstrated in BRECQ. However, it is a challenge for OS+ and AWQ to expand to block-wise quantization, which would significantly enlarge the solution space and increase the time of grid searching.
>
> |      | operation     | position                 | optimization                                          |
> | ---- | ------------- | ------------------------ | ----------------------------------------------------- |
> | AWQ  | scale         | linear layer             | grid searching                                        |
> | OS+  | scale & shift | linear layer             | grid searching for scaling & pre-defined for shifting |
> | LET  | scale & shift | linear layer & attention | gradient decent                                       |
>
> We have added this discussion in Section A2 of our revision.  In Figure A3 of Appendix, we also compare the quantization error of each block. We can find that OmniQuant significantly reduce the quantization loss compared with the grid-searching based method such as AWQ and Outlier Suppression +. You can also refer [Genera Response](https://openreview.net/forum?id=8Wuvhh0LYW&noteId=D6esYYnEi2) for more discussion about the novelty of our paper.
>
>
>
> **Q4:** Experiments shall be compared with the paper Outlier Suppression+ because you and they work on the same quantization problem, take the same equivalent transformation, and have similar optimization designs.
>
> **A4:**  We have incorporated comparisons with OS+ in Table 2 of our manuscript, and offered an overview of the W4A4 quantization with six zero-shot tasks as follows. We can find that OmniQuant outperforms OS+ in a range from 4.01% to 6.68% accuracy.
>
> | W4A4        | Quantization | Average Acc. |
> | ----------- | ------------ | ------------ |
> | LLaMa-1-7B  | OS+          | 48.43        |
> | LLaMa-1-7B  | OmniQuant    | 52.65        |
> | LLaMa-1-13B | OS+          | 49.86        |
> | LLaMa-1-13B | OmniQuant    | 54.37        |
> | LLaMa-1-30B | OS+          | 52.62        |
> | LLaMa-1-30B | OmniQuant    | 56.63        |
> | LLaMa-1-65B | OS+          | 52.52        |
> | LLaMa-1-65B | OmniQuant    | 59.23        |

---

> ### Author Response · Authors · 2023-11-17
> **Response to Reviewer WDuc (Part: 3/3)**
>
> **Q5:** I noticed that the paper requires careful equivalent parameter initialization via the compared baseline SmoothQuant. I might wonder how it behaves without good initialization. For example, under asymmetric cases W4A8, W4A6, and W8A4, where the LLM-QAT shows SmoothQuant can behave terribly.
>
> **A5:** Good advice, sorry for missing this ablation. To validate the impact of initialization, we first try to initial scaling as 1 and initial shifting as 0. We do experiments on LLaMa-1-7B with W4A4 quantization, and report the average perplexity and average accuracy. As shown in the following table, we can find that both careful initialization of scaling and shifting can improve the final performance. Specifically, scaling initialization is more important than shifting due to that scaling plays the main role in alleviating outliers.
>
> | LLaMa-1-7B W4A4          | Average PPL | Average Acc. |
> | ------------------------ | ----------- | ------------ |
> | careful initialization   | 12.87       | 52.65        |
> | initialize scaling as 1  | 13.64       | 51.37        |
> | initialize shifting as 0 | 12.95       | 52.22        |
>
> Moreover, as shown in the following table, OmniQuant also demonstrates robust performance in asymmetric cases, ensuring effectiveness across various scenarios.
>
> | LLaMa-1-7B | Average PPL | Average Acc. |
> | ---------- | ----------- | ------------ |
> | FP         | 6.38        | 64.09        |
> | W4A4       | 12.87       | 52.65        |
> | W4A8       | 6.60        | 62.40        |
> | W4A6       | 6.85        | 61.64        |
> | W8A4       | 11.52       | 53.46        |
>
> We have added these experiments in Table A6 and Table A17 of our revision.
>
>
>
> **Q6:** I see the challenge of the two techniques is how to combine the two kinds of learning together as they influence each other. However, current techniques seem can not solve this problem well. Thus, I think it would be better if the paper gives more description, and design consideration to the combination part, which might increase the novelty. For example, maybe alternately train these two techniques. For example, maybe alternately train these two techniques.
>
> **A6:** Thanks for the suggestion. We want to emphasize that in our approach, LWC and LET are trained simultaneously, and we have also explored an iterative training approach by iterations or epochs. The results, as presented in the following table, clearly indicate that training LWC and LET simultaneously yields the best performance. As aforementioned in Q2, LWC leverages the extremums after LET as the reference, allowing it to capture the dynamic changes in the weight range introduced by LET. This synergy between LET and LWC creates a progressive process, where both techniques reinforce each other rather than interfere. To further support this statement, we conducted an additional experiment (last row in the table), training LWC and LET iteratively with double the epochs. The results show that simultaneous training with 20 epochs achieves comparable performance to iterative training with 40 epochs. This demonstrates the effectiveness and efficiency of training LWC and LET simultaneously. We have added this experiment in Table A8 of our revision.
>
> | LLaMa-1-7B/ W4A4/ Iterative LWC and LET | Average PPL | Average Acc. |
> | --------------------------------------- | ----------- | ------------ |
> | simultaneous training (OmniQuant)       | 12.87       | 52.65        |
> | each iteration                          | 13.56       | 50.91        |
> | each epoch                              | 13.51       | 52.06        |
> | each epoch + double training epochs     | 12.80       | 52.50        |
>
>
>
> We sincerely appreciate the time and efforts you have dedicated to reviewing our paper. Should you have any further inquiries, please let us know and we are more than delighted to discuss with you and run more experiments for any pieces of your interests in our work.

---

> ### Author Response · Authors · 2023-11-20
> **Look forward to further discussion.**
>
> Dear Reviewer WDuc,
>
> We hope that our response can address your concerns. As the deadline for discussion period is approaching, we really appreciate if you can let us know whether there still exists any further question about the paper or the response. We are looking forward to further discussion.
>
> Best regards,
>
> Authors of Paper 1585

---

> ### Comment · Reviewer_WDuc · 2023-11-20
> **Post-rebuttal**
>
> Thanks for your detailed reply. Some of my concerns have been addressed. I'd like to increase my score to 5. However, for the distinction compared to AWQ and Outlier Suppression+, I knew the difference but as stated in my first review, what I wonder is why the optimization method proposed in this paper is more beneficial. For example, the shifting technique is first proposed by Outlier Suppression+, though they use a pre-defined way, the paper seems not to show the results that learning the shifting vector can help a lot. Also, the table above shows the results of OmniQuant which takes both the LET and weight clipping. However, it did not show how much benefit can the gradient descent of LET bring compared to that of Outlier Suppression+ as that paper demonstrated lower accuracy obtained by learning. I think detailed analyses are very important as the difference is not big and novelty is limited in this part.

---

> > ### Author Response · Authors · 2023-11-22
> > **More comparisons between LET and Outlier Suppression+**
> >
> > Dear Reviewer WDuc,
> >
> > Thank you for your feedback!  We appreciate the opportunity to address your concerns and provide a more in-depth comparison between LET and Outlier Suppression+ (OS+).
> >
> > To enhance clarity, we present a comprehensive comparative study in the table below, highlighting the distinct advantages of learnable scaling and shifting within our LET framework compared to their predefined counterparts in SmoothQuant and OS+:
> >
> > | LLaMa-1-7B W4A4                                    | Average PPL | Average Acc. |
> > | -------------------------------------------------- | ----------- | ------------ |
> > | predefined scaling (Smoothquant)                   | 28.78       | 38.41        |
> > | learnable scaling                                  | 17.52       | 48.30        |
> > | learnable scaling + predefined shifting            | 17.60       | 48.10        |
> > | learnable scaling + learnable shifting （Our LET） | 16.97       | 48.84        |
> >
> > These results clearly demonstrate that learnable scaling significantly enhances performance when compared to its predefined counterpart. Furthermore, the introduction of learnable shifting proves beneficial to learnable scaling, an advantage not observed with predefined shifting. This underscores the flexibility of our approach, as learnable components can adapt to the current distribution dynamically.
> >
> > In addition to intra-framework analysis, we conducted comparisons between LET and OS+, detailed in the following table. The results consistently demonstrate LET's superior performance across both LLaMa and OPT model families. Particularly noteworthy is LET's significant advantage in OPT, where the activation outlier challenge is more pronounced [1], rendering grid-searched optimization approaches like OS+ less effective. Furthermore, LET's compatibility with other differential methods, such as LWC, showcases its potential for further performance improvements.
> >
> > | W4A4        | Quantization | Average Acc. |
> > | ----------- | ------------ | ------------ |
> > | LLaMa-1-7B  | OS+          | 48.43        |
> > | LLaMa-1-7B  | LET          | 48.83        |
> > | LLaMa-1-7B  | LET+LWC      | 52.65        |
> > | LLaMa-1-13B | OS+          | 49.86        |
> > | LLaMa-1-13B | LET          | 50.81        |
> > | LLaMa-1-13B | LET + LWC    | 54.37        |
> > | OPT-6.7B    | OS+          | 42.57        |
> > | OPT-6.7B    | LET          | 51.14        |
> > | OPT-6.7B    | LET + LWC    | 58.29        |
> > | OPT-13B     | OS+          | 43.62        |
> > | OPT-13B     | LET          | 51.67        |
> > | OPT-13B     | LET + LWC    | 59.03        |
> >
> > In summary, our extended analysis reinforces the efficacy of LET in comparison to existing methods, showcasing its adaptability, superior performance, and seamless integration with other differential techniques.
> >
> > We appreciate your time and consideration and welcome any further feedback you may have.
> >
> > [1] Enhancing Computation Efficiency in Large Language Models through Weight and Activation Quantization

---

> > > ### Comment · Reviewer_WDuc · 2023-11-23
> > > **Post-rebuttal**
> > >
> > > Thank you for the reply. I'd like to increase my score to 6. Besides, I wonder that as the activation outlier challenge is more pronounced in OPT, why this method behaves better on OPT compared to LLaMA. Also, I notice that the 4-bit OPT results in the appendix can reach the FP baseline while the 4-bit LLaMA still suffers a large accuracy drop. Could the author provide further explanation for this phenomenon?

---

> > > > ### Author Response · Authors · 2023-11-23
> > > > **In-depth exploration about the outlier patterns in OPT and LLaMa families**
> > > >
> > > > Dear Reviewer WDuc,
> > > >
> > > > Thank you for your feedback. We appreciate to give a more In-depth exploration and discussion about why OPT family suffer less performance degeneration compared with LLaMa family and can even achieve the  FP baseline with W4A4 quantization.
> > > >
> > > > We address this matter through two perspectives:
> > > >
> > > > 1. **The more pronounced outliers in OPT lead grid-searched optimization approaches like OS+  can not find appropriate solution.**
> > > >
> > > > To assess the challenge posed by outliers, we calculated the average values of each channel, with larger magnitudes indicating significant outliers. Taking the first normalization layer in the 8th block as an example, we present the top-10 average values for OPT-6.7B and LLaMa-7B in the table below. Notably, both models exhibit outlier activation channels, but the magnitude of outliers is larger in OPT compared to LLaMa (53.9538 vs. 5.6836). **This trend is consistent across all other blocks.**
> > > >
> > > > | Model    | Top-10 values                                                |
> > > > | -------- | ------------------------------------------------------------ |
> > > > | OPT-6.7B | **53.5938**,  9.2422,  7.2812,  3.0762,  2.1699,  1.3447,  1.2949,  1.2305, 1.1191,  0.9805 |
> > > > | LLaMa-7B | **5.6836**, 3.3945, 2.6113, 2.4473, 2.3594, 2.3281, 1.6250, 1.5723, 1.5039,1.4844 |
> > > >
> > > > The presence of more significant outliers in OPT contributes to greater quantization loss with Round-to-Nearest (RTN) activation quantization, resulting in the observed performance degeneration in the table below.
> > > >
> > > > | Average PPL | OPT-6.7B | LLaMa-7B |
> > > > | ----------- | -------- | -------- |
> > > > | FP (16bit)  | 11.29    | 6.38     |
> > > > | W4A16-RTN   | 12.74    | 6.84     |
> > > > | W16A4-RTN   | 2.8 e5   | 273.20   |
> > > > | W4A4-RTN    | 2.9 e5   | 524.65   |
> > > >
> > > > Therefore, these pronounced outliers pose optimization challenges, rendering grid-searched optimization approaches like OS+ ineffective for OPT models. However, our proposed LET successfully navigates these challenges, finding optimal solutions for both OPT and LLaMa models through differentiable optimization methods.
> > > >
> > > > 2. **Consistency of outlier magnitude in OPT across different samples facilitates channel-wise operations.**
> > > >
> > > > The outliers in OPT consistently exhibit the similar magnitude across different samples, while the outliers in LLaMa vary. This consistent regulation of outliers in OPT makes it more suitable for channel-wise operations, such as LET, and suggests the potential to achieve FP-16 performance with W4A4 through OmniQuant. To illustrate, we selected the first five samples from WikiText, calculated the average output values of each channel in the first normalization layer of the 8-th block, and reported the channel indices of the top-5 values. The tables below show that OPT-6.7B maintains the almost same index order across different samples, whereas LLaMa-7B shows considerable variation. **Note that we did not cherry-pick the results, and such phenomena can be easily observed in different blocks and samples.**
> > > >
> > > > | Model    | Sample Index | Top-5 channel index          |
> > > > | -------- | ------------ | ---------------------------- |
> > > > | OPT-6.7B | Sample 1     | 2394, 2972, 3136, 1978, 1778 |
> > > > | OPT-6.7B | Sample 2     | 2394, 2972, 3136, 1978, 1778 |
> > > > | OPT-6.7B | Sample 3     | 2394, 2972, 3136, 1978, 1666 |
> > > > | OPT-6.7B | Sample 4     | 2394, 2972, 3136, 1978, 1778 |
> > > > | OPT-6.7B | Sample 5     | 2394, 2972, 3136, 1978, 1778 |
> > > > | LLaMa-7B | Sample 1     | 704, 3840, 2877,  830,  690  |
> > > > | LLaMa-7B | Sample 2     | 2877,  487,  690, 3840,  830 |
> > > > | LLaMa-7B | Sample 3     | 487, 2877,  690,  830, 2172  |
> > > > | LLaMa-7B | Sample 4     | 2877,  690, 2172,  487,  830 |
> > > > | LLaMa-7B | Sample 5     | 3840,  487, 2877,  777,  690 |
> > > >
> > > > In summary, our extended analysis provides a more in-depth understanding of the differing outlier patterns in OPT and LLaMa models. While OPT models are well-suited for channel-wise operations due to the consistent distribution of outlier channels across different samples, they also face optimization challenges due to the significant magnitude of outliers. Fortunately, our proposed OmniQuant effectively mitigates these optimization issues and leverages the pattern of OPT outliers to achieve superior performance.
> > > >
> > > > We appreciate your time and insightful consideration, and welcome any further feedback you may have.

---

### Official Review · Reviewer_ZyAK · 2023-11-01

**Soundness:** 3 good
**Presentation:** 3 good
**Contribution:** 3 good
**Rating:** 6
**Confidence:** 5

**Summary:**

This paper addresses the challenges faced by large language models (LLMs) by optimizing quantization parameters. It is based on the SmoothQuant and Outlier Suppression+ and mainly contributes to a learnable pipeline. The idea is simple and trivial. However, the effect is good on various models and datasets.

**Strengths:**

- Extend the existing quantization methods based on the thought of transformation to a learnable one, and give a pipeline with a stable optimization process
- the idea of learnable scaling is simple, but making the learning stable and effective is a good contribution
- conduct experiments on various models

**Weaknesses:**

- The novelty is limited. The overall framework is based on two existing methods.
- The learnable idea is not new. In Outlier Suppression+, the scaling has been designed to be learned via a scheme that does not depend on gradient.

**Questions:**

- Both outlier suppression+ and this paper highlight the scaling to be learned. An in-depth comparison needs to be provided, including the experimental perspective and the theoretical perspective.
- The optimization based on little data and backward propagation makes the learning easy to be overfitted. More validation should be conducted to prove the generalization ability of this learning.
- There are some new kinds of ways to decompose the outliers, e.g., https://arxiv.org/abs/2310.08041. Comprehensive experiments are suggested to further enrich the validation.

---

> ### Author Response · Authors · 2023-11-17
> **Response to Reviewer ZyAK (Part: 1/2)**
>
> We thank the Review ZyAK for the time and effort to review our paper. We are glad that the reviewer found our contribution to constructing the gradient optimization framework. We have summarized the updating of our revision in [Summary of Paper Updating](https://openreview.net/forum?id=8Wuvhh0LYW&noteId=YV1oX701CV) and provided an in-depth discussion about the novelty of our method in [Genera Response](https://openreview.net/forum?id=8Wuvhh0LYW&noteId=D6esYYnEi2). Below, we address each identified weakness.
>
>
>
> **Q1**: The novelty is limited. The overall framework is based on two existing methods.
>
> **A1:** Thanks for the comment. We argue that although optimally determining clipping threshold and equivalent parameters have been explored in existing literature, our proposed OmniQuant is still novel enough by the optimal design. We clarify our novelty from $4$ aspects. 1) the first differentiable quantization framework for LLMs; 2) the novelty of LET; 3) the novelty of LWC; 4) the seamless combination of LWC and LET. Please refer [General Response](https://openreview.net/forum?id=8Wuvhh0LYW&noteId=D6esYYnEi2) for more details.
>
>
>
> **Q2:** The learnable idea is not new. In Outlier Suppression+, the scaling has been designed to be learned via a scheme that does not depend on gradient.
>
> **A2:** Respecting your observation, we highlight that the novelty of our learnable approach lies in differential gradient updates for various quantization parameters. As illustrated in Figure A3, OmniQuant's differential gradient-based learning significantly reduces the quantization error compared with the grid-searching method employed in Outlier Suppression+. In experiments, Table 2 in the paper demonstrates the superiority of OmniQuant to OS+.
>
>
>
> **Q3:** An in-depth comparison needs to be provided, including the experimental perspective and the theoretical perspective.
>
> **A3:**  OmniQuant differs from Outlier Suppression+ in several aspects, including parameter optimization through gradient updates, block-wise optimization, expansion of equivalent transformation to attention operations, and the introduction of learnable weight clipping (LWC). Details are as follows.
>
> - Both OmniQuant and OS+ try to find the optimal equivalent transformation parameters. OS+ solves this problem by pre-defined shifting strength and grid-searched scaling parameters. However, OmniQuant obtains both shifting and scaling parameters through gradient optimization.
>
> - OmniQuant leverages block-wise optimization while OS+ executes grid searching through a mix of single linear layer and multiple linear layers objective(Eq.6 and Eq.7 in their paper). Block-wise optimization considers interaction within a block and can produce better performance as demonstrated in BRECQ. However, it is a challenge for OS+ to expand to block-wise quantization, which would significantly enlarge the solution space and increase the time of grid searching.
>
> - LET expends the equivalent transformation to attention operation (Eq. (5) in our paper), but OS+ only does the equivalent transformation on linear layers.
>
> - OmniQuant also introduces learnable weight clipping (LWC), which further facilitates the low-bits (2,3) weight-only quantization, while Outlier Suppression only supports for weight-activation quantization.
>
> We also incorporated a comparison with Outlier Suppression+ in Table 2 . An overview is as follows.
>
> | W4A4        | Quantization | Average Acc. |
> | ----------- | ------------ | ------------ |
> | LLaMa-1-7B  | OS+          | 48.43        |
> | LLaMa-1-7B  | OmniQuant    | 52.65        |
> | LLaMa-1-13B | OS+          | 49.86        |
> | LLaMa-1-13B | OmniQuant    | 54.37        |
> | LLaMa-1-30B | OS+          | 52.62        |
> | LLaMa-1-30B | OmniQuant    | 56.63        |
> | LLaMa-1-65B | OS+          | 52.52        |
> | LLaMa-1-65B | OmniQuant    | 59.23        |

---

> ### Author Response · Authors · 2023-11-17
> **Response to Reviewer ZyAK (Part: 2/2)**
>
> **Q4:** The optimization based on little data and backward propagation makes the learning easy to be overfitted. More validation should be conducted to prove the generalization ability of this learning.
>
> **A4:** Acknowledging the potential for overfitting, we highlight that OmniQuant mitigates this issue through a constrained solution space. Specifically, optimization focuses on LET and LWC rather than all weights. Our results in Table A11 of the Appendix demonstrate OmniQuant's superior performance even with only 16 calibration samples. Additionally, we present performance on MMLU to further showcase the generalization ability of our approach.
>
> | LLaMa-1-7B (FP: 38.41 ) | W4A16g128 | W3A16g128 | W2A16g128 | W4A4  |
> | ----------------------- | --------- | --------- | --------- | ----- |
> | RTN                     | 37.37     | 33.43     | 22.55     | 23.31 |
> | GPTQ                    | 35.39     | 30.53     | 23.83     | -     |
> | AWQ                     | 37.71     | 35.43     | 22.58     | -     |
> | Outlier Suppression+    | -         | -         | -         | 25.72 |
> | OmniQuant               | 37.50     | 35.60     | 26.03     | 26.93 |
>
> **Q5:** There are some new kinds of ways to decompose the outliers, e.g., https://arxiv.org/abs/2310.08041. Comprehensive experiments are suggested to further enrich the validation.
>
> **A5:** Thanks for your suggestion. Your referred paper QLLM (https://arxiv.org/abs/2310.08041) alleviates outlier channels by disassembling instead of scaling. In my view, Outier disassembling only considers the outlier channels. However, LET in our paper, as shown in Figure A2, can also adjust the range of normal channels to achieve a more quantization-friendly distribution. Therefore, the combined disassembling operation of QLLM and LET of our paper may further boost the performance of quantization models.  We will undertake this exploration after QLLM releases its code and engage in discussions with the QLLM team for further insights.
>
> We sincerely appreciate the time and efforts you have dedicated to reviewing our paper. Should you have any further inquiries, please let us know and we are more than delighted to discuss with you.
>
> [1] Brecq: Pushing the limit of post-training quantization by block reconstruction

---

> > ### Comment · Reviewer_ZyAK · 2023-12-03
> >
> > Thanks for your detailed reply. Most of my concerns have been addressed. However, the overfitting problem will somehow limit the upper bound of the learning-based method. Suggest that more evidence to prove the generalization ability can be included in the final version.

---

> ### Author Response · Authors · 2023-11-22
> **Last day reminder and looking forward to discussion**
>
> Dear reviewer ZyAK,
>
> Thanks again for your valuable time and insightful comments. As the deadline for the Author/Reviewer discussion is approaching, it would be nice of you to let us know whether our answers have solved your concerns so that we can better improve our work. We are happy to provide any additional clarifications that you may need.
>
> Best regards!

---

### Official Review · Reviewer_FuSa · 2023-11-01

**Soundness:** 3 good
**Presentation:** 3 good
**Contribution:** 3 good
**Rating:** 6
**Confidence:** 4

**Summary:**

This paper proposes OmniQuant,  a novel quantization technique for large language models (LLMs). OmniQuant introduces two learnable approaches to calibrate the quantized model, which are Learnable Weight Clipping (LWC) and Learnable Equivalent Transformation (LET). The calibration is conducted in a block-wise manner and uses gradient updates to minimize the quantization error. The paper evaluates OmniQuant on various LLMs, quantization configurations, and natural language tasks.

**Strengths:**

- The LWC method proposed in the paper is simple yet effective, outperforming previous clipping-based approaches. The LET method addresses the shortcomings of SmoothQuant and contributes to better activation quantization performance.
- The OmniQuant framework can be applied to both weight-only quantization and weight & activation quantization. The calibration process is relatively simple and fast.
- Comprehensive ablation studies are conducted to analyze the effectiveness of each proposed technique.

**Weaknesses:**

- It would be beneficial to have additional experiments on more complex tasks. I am wondering how OmniQuant impacts the reasoning ability of LLMs, which can be evaluated by MMLU. GPT-4 evaluation is a bit ad-hoc nowadays, and there are also several better benchmarks to measure the instruction-tuned models performance, such as MT-Bench or AlpacaEval (correction: should be AlpacaEval instead of AlpacaFarm). Evaluating some stronger chatbots like Vicuna-v1.5 on them should be conducted.
- Some strong related work is not discussed or compared, such as SpQR [1] and SqueezeLLM [2]. For instance, SqueezeLLM outperforms the proposed approach for wiki and c4 perplexity on LLaMA v1 7b and 13b under 3-bit and 4-bit weight-only quantization settings (see table 1 in their paper). Additional discussion and results should be added to compare OmniQuant with them.

[1] Dettmers, T., Svirschevski, R., Egiazarian, V., Kuznedelev, D., Frantar, E., Ashkboos, S., Borzunov, A., Hoefler, T. and Alistarh, D., 2023. SpQR: A Sparse-Quantized Representation for Near-Lossless LLM Weight Compression. arXiv preprint arXiv:2306.03078.
[2] Kim, S., Hooper, C., Gholami, A., Dong, Z., Li, X., Shen, S., Mahoney, M.W. and Keutzer, K., 2023. SqueezeLLM: Dense-and-Sparse Quantization. arXiv preprint arXiv:2306.07629.

**Questions:**

Please address the weaknesses mentioned above.

---

> ### Author Response · Authors · 2023-11-17
> **Response to Reviewer FuSa**
>
> Thanks for your thoughtful review that will help us strengthen the manuscript. We have summarized the updating of our revision in [Summary of Paper Updating](https://openreview.net/forum?id=8Wuvhh0LYW&noteId=YV1oX701CV). Below, we address each identified weakness.
>
> **Q1**: Results on MMLU.
>
> **A1:** As shown in the following table, we report the average accuracy on MMLU, showcasing that OmniQuant consistently achieves comparable or superior performance in reasoning ability. We have added this experiment in Table A16 of our revision.
>
> | LLaMa-1-7B (FP: 38.41 ) | W4A16g128 | W3A16g128 | W2A16g128 | W4A4  |
> | ----------------------- | --------- | --------- | --------- | ----- |
> | RTN                     | 37.37     | 33.43     | 22.55     | 23.31 |
> | GPTQ                    | 35.39     | 30.53     | 23.83     | -     |
> | AWQ                     | 37.71     | 35.43     | 22.58     | -     |
> | Outlier Suppression+    | -         | -         | -         | 25.72 |
> | OmniQuant               | 37.50     | 35.60     | 26.03     | 26.93 |
>
>
>
> **Q2:**  Evaluating instruction-tuned models on MT-Bench or AlpacaFarm and evaluating stronger Chatbot.
>
> **A2:** We have quantized Vicuna-v1.5 using W3A16g128 quantization and compared the performance of different methods through MT-Bench. The results reveal that OmniQuant consistently outperforms RTN and AWQ.
>
> | Vicuna-v1.5-7B-W3A16g128 | OmniQuant Win | Tie  | OmniQuant Lost | Win rate |
> | ------------------------ | ------------- | ---- | -------------- | -------- |
> | Omniquant v.s. RTN       | 101           | 239  | 20             | 83.4 %   |
> | Omniquant v.s. AWQ       | 70            | 225  | 65             | 51.8%    |
>
>
>
> **Q3:** Discussion and comparison with related work- SpQR and SqueezeLLM
>
> **A3:** Thanks for your suggestion, we have included the discussion and comparison with SqQR and SqueezeLLM in Section A7. However, we would like to emphasize two crucial points. Firstly, SpQR and SqueezeLLM are concurrent works, both available on arXiv in June 2023, which falls within four months of the ICLR submission deadline (Please refer to the policy in the last Q&A at https://iclr.cc/Conferences/2024/ReviewerGuide ). Secondly, Although OmniQuant may perform slightly worse than SqueezeLLM, our OmniQuant focuses on uniform (INT) quantization which offers simplicity and flexibility and has strong hard-ware support. Moreover, OmniQuant achieves competitive performance in both weight-only quantization and weight-activation quantization. In contrast, SpQR and SqueezeLLM only support weight-only quantization. We believe this distinction adds valuable context to the comparison.
>
>
>
> We sincerely appreciate the time and efforts you have dedicated to reviewing our paper. Should you have any further inquiries, please let us know and we are more than delighted to discuss with you and run more experiments for any pieces of your interests in our work.
>
>
>
> [1] Judging LLM-as-a-Judge with MT-Bench and Chatbot Arena

---

> ### Author Response · Authors · 2023-11-22
> **Last day reminder and looking forward to discussion**
>
> Dear reviewer FuSa,
>
> Thanks again for your valuable time and insightful comments. As the deadline for the Author/Reviewer discussion is approaching, it would be nice of you to let us know whether our answers have solved your concerns so that we can better improve our work. We are happy to provide any additional clarifications that you may need.
>
> Best regards!

---

> > ### Comment · Reviewer_FuSa · 2023-11-22
> > **Response**
> >
> > Dear Authors,
> >
> > Thank you for the comprehensive rebuttal and the clarifications provided. I have also read other reviews and rebuttals. At present, I have no further questions and would like to keep my current positive rating.

---

> > > ### Author Response · Authors · 2023-11-23
> > > **Thanks for your feedback.**
> > >
> > > Dear Reviewer FuSa,
> > >
> > > Thank you for your feedback! We appreciate the constructive reviews for improving our work.
> > >
> > > Best regards,
> > >
> > > Authors of Paper 1585

---

### Author Response · Authors · 2023-11-17
**Summary of Paper Updating**

We thank all the reviewers for their valuable feedback and great efforts, which have significantly contributed to the enhancement of our paper. We have diligently addressed all comments, questions, and concerns, with major modifications highlighted in blue within the attached PDF. Summarized major changes in our updated version:

1) **We added more comparisons and discussion**

-  Added comparisons with Outlier Suppression+ (OS+) in Table 2.-  Provided a detailed discussion on the distinctions from existing equivalent transformation methods, including Smoothquang, AWQ and Outlier Suppression + (see Section A2 in Appendix).

-  Added the comparison of quantization error among OmniQuant, AWQ and Outlier Suppression +. (See Figure A3 in Appendix)

-  Added discussion and comparison about SpQR and SqueezeLLM. (see Section A7 in Appendix)

-  Presented MMLU results in Table A16.

2) **We added more ablation studies.**

- Exploded the combination with other equivalent transformation methods or weight clipping method. (see Table A2 in Appendix)

- Included an ablation study on the effect of LET position (see Table A5 in Appendix).

- Conducted an ablation study on the initialization of LET parameters (see Table A6 in Appendix).

 - Explored the impact of the quantization of Softmax output. (see Table A7 in Appendix)

 - Explored the effect of iterative training. (see Table A8 in Appendix)

3) **Addressed other phrasing clarifications requested by reviewers.**

---

### Author Response · Authors · 2023-11-17
**General response (About novelty of our method)**

We thank all the reviewers for their suggestions. To address the reviewers' concerns about our paper which mainly lie in the novelty of our method, we provide an in-depth discussion and summarize the novelty of OmniQuant as follows:

1) **OmniQuant is the first differentiable post-training quantization algorithm for large language models.** While differentiability is straightforward for typical CNNs and ViTs, the numerous parameters of large language models lead to efficiency challenges. Based on block-wise optimization, we make it possible to differentiably quantize LLMs. We also design learnable weight clipping (LWC) and learnable equivalent transformation (LET) to achieve competitive performance for the quantized models.

2. **Novelty of LET**: Previous equivalent transformation methods, such as Outlier Supression+ (OS+), only consider the equivalent transformation of linear layers. LET firstly expend equivalent transformation to matrix multiplication within attention (Eq. (5) in our paper). Then, all LET parameters are determined through gradient descent. Although OS+ has undertaken a similar pattern, it relies on hierarchical rules and optimization with grid-search parameters, resulting in a larger quantization error than our OmniQuant. as shown in Fig. A3 of the revision. Moreover, comparative experiments in Table 2 validate the superiority of OmniQuant to OS+.

3. **Novelty of LWC**: Despite learning clipping threshold is not new, we re-parameterize the clipping threshold by learnable clipping strengths and the extremums of weights altered by LET.  Such a design helps LWC to capture the dramatic weight change caused by LET with reference to the extremums of the current weights, reducing the optimization difficulty. During the rebuttal, we demonstrate this insight in Fig. A5 of the revision. In experiments, OmniQuant with LWC achieves superior performance compared with OmniQuant with PACT and LSQ as shown in Table A14 of the paper.

4. **The synergy between LET and LWC is achieved through a sophisticated differentiable framework as demonstrated in Algorithm 1**, not a simple additive combination. LET performs activation-to-weight migration, and LWC further facilitates the quantization of weights, resulting in a seamless integration of the two techniques.  *More ablation studies are added to validate the effectiveness of the combination of LWC and LET.* To demonstrate the effect of each proposed component and the advantage of combining LET and LWC, We also test other combination variants, including replacing LET with SmoothQuant or replacing LWC with grid-searched weight clipping. The results show that the combination of the proposed LET and LWC  achieves the best performance.


| LLaMa-1-7B W4A4        | Average PPL | Average Acc. |
| ---------------------- | ----------- | ------------ |
| SmoothQuant            | 28.78       | 38.41        |
| LET                    | 16.97       | 48.83        |
| LET + grid-searched WC | 15.82       | 49.59        |
| SmoothQuant + LWC      | 15.80       | 50.15        |
| **LET+LWC**            | **12.87**   | **52.65**    |

---

### Meta-Review · Area_Chair_MPtF · 2023-12-02

**Metareview:**

All reviewers find the papers contributions substantial and the proposed method simple and effective. Main concerns were around missing comparisons to some recent works. Authors have addressed this in the rebuttal presenting additional results. All reviewers suggest acceptance and I am happy to recommend acceptance for this paper.

**Justification For Why Not Higher Score:**

Limited novelty and technical contributions.

**Justification For Why Not Lower Score:**

simple but effective approach with through comparisons.

---

### Decision · Program_Chairs · 2024-01-16

Accept (spotlight)